
# Ice water path retrievals from Meteosat-9 using quantile regression neural networks

Adrià Amell, Patrick Eriksson, and Simon Pfreundschuh

Department of Space, Earth and Environment, Chalmers University of Technology, Gothenburg, Sweden

**Correspondence:** Adrià Amell (amell@chalmers.se)

**Abstract.** The relationship between geostationary radiances and ice water path (IWP) is complex, and traditional retrieval approaches are not optimal. This work applies machine learning to improve the IWP retrieval from Meteosat-9 observations, with a focus on low latitudes, training the models against retrievals based on CloudSat. Advantages of machine learning include avoiding explicit physical assumptions on the data, an efficient use of information from all channels, and easily leveraging

spatial information.

Thermal infrared (IR) retrievals are used as input to achieve a performance independent of the solar angle. They are compared with retrievals including solar reflectances, as well as a subset of IR channels for compatibility with historical sensors. The retrievals are accomplished with quantile regression neural networks. This network type provides case-specific uncertainty estimates, compatible with non-Gaussian errors, and is flexible enough to be applied to different network architectures.

Spatial information is incorporated into the network through a convolutional neural network (CNN) architecture. This choice outperforms architectures that only work pixelwise. In fact, the CNN shows a good retrieval performance by using only IR channels. This allows computing diurnal cycles, a problem that CloudSat cannot resolve due to its limited temporal and spatial sampling. These retrievals compare favourably with IWP retrievals in CLAAS, a thoroughly validated dataset based on a traditional approach. These results highlight the possibilities to overcome limitations from physics-based approaches using

machine learning while providing efficient, probabilistic IWP retrieval methods. Moreover, they suggest extending this first work to higher latitudes as well as considering geostationary data as a complement to the upcoming Ice Cloud Imager mission, for example, to bridge the gap in temporal sampling with respect to space-based radars.

## 1   Introduction

Clouds remain among the main factors that hinder climate models to give a confident value for the climate sensitivity. According

to the Sixth Assessment Report, the last report from the Intergovernmental Panel on Climate Change (IPCC, 2021), there is now high confidence in the feedbacks associated with the subtropical marine low-cloud regime and altitude of high clouds. There has been less progress on the tropical high-cloud amount feedback and this component is the largest contributor to the overall cloud feedback uncertainty (Forster et al., 2021). That said, global warming is ongoing and will continue. One of the critical aspects of this warming is changes in precipitation, that also are difficult to predict with accuracy. A quantity with

relations to both these modelling challenges is the mass of ice hydrometeors. At low ice concentrations, the radiative forcing of



ice clouds follows the ice water content (IWC, $\mathrm{kg\,m^{-3}}$), albeit altitude and time of day must also be considered. Precipitation in the form of snow, graupel and hail is directly linked to the masses of ice hydrometeors inside the atmosphere, but also rain is affected by the ice water contents above.

The total amount of ice hydrometeors is normally reported as the ice water path (IWP, $\mathrm{kg\,m^{-2}}$), which is one of the essential climate variables from the Global Climate Observing System (GCOS, 2021). Despite it is an integrated value and should be easier to constrain than level-specific IWC, there has been little progress around IWP in both measurements and models (Waliser et al., 2009; Eliasson et al., 2011; Duncan and Eriksson, 2018). The most accurate global data on IWP should be provided by retrievals based on CloudSat reflectivites such as DARDAR (Cazenave et al., 2019). Where the CloudSat $94\,\mathrm{GHz}$ radar measures it gives accurate information at high vertical resolution, except for high IWP values where attenuation and multiple scattering decrease the retrieval accuracy. However, the swath width of CloudSat is only $1.4\,\mathrm{km}$.

Passive instruments offer a much higher horizontal coverage. Therefore, passive observations are a good complement to CloudSat. In 2025 the Ice Cloud Imager (ICI) will be launched aboard Metop Second Generation (Metop-SG) B and will provide information on IWP over a swath $1\,500\,\mathrm{km}$ wide with a $16\,\mathrm{km}$ horizontal resolution (Eriksson et al., 2020). That is, the geographical coverage of the ICI radiometer is more than a factor 100 higher than CloudSat, and close to global coverage is obtained on a daily basis. The ICI IWP retrieval accuracy should be comparable to the one allowed by a $94\,\mathrm{GHz}$ radar (Pfreundschuh et al., 2020) and retrievals of coarse IWC profiles should be possible (Brath et al., 2018), but cloud radars are still far superior in terms of spatial resolution.

Microwave instruments are so far only operated on satellites in low orbits. Observations from geostationary satellites are an important complement as they provide short revisit times. For instance, the Spinning Enhanced Visible and InfraRed Imager (SEVIRI) instrument aboard the Meteosat Second Generation (MSG) of geostationary satellites (Schmetz et al., 2002) provides full disc images every 15 minutes. The main challenge with retrievals from geostationary satellites is the complex relationship between visible and infrared (VISIR) radiances and IWP.

Nakajima and King (1990) found that the reflectances at $0.75\,\text{μm}$ are primarily sensitive to cloud optical thickness $\tau$, while reflectances at $2.16\,\text{μm}$ are primarily sensitive to effective droplet radius $r_e$. Assuming a sufficiently representative $r_e$ value, then IWP can be estimated from these two parameters (Stephens, 1978). This solar bispectral method constitutes the foundation of several IWP retrieval methods based on VISIR radiances. This includes: the Cloud Physical Properties algorithm (CPP, first published by Roebeling et al., 2006), the Daytime Cloud Optical and Microphysical Properties (DCOMP) algorithm used in Pathfinder Atmospheric Extended (PATMOS-X, Walther and Heidinger, 2012), the Moderate Resolution Imaging Spectroradiometer (MODIS) cloud properties product (Platnick et al., 2017) or the NASA Clouds and the Earth's Radiant Energy System (CERES) project algorithms (Minnis et al., 2011, 2021), the latter originally developed for MODIS but also adapted for other polar-orbiting (Minnis et al., 2016a) and geostationary imagers (Minnis et al., 2008) in the Satellite Cloud and Radiation Property retrieval System (SatCORPS). All these retrieval algorithms estimate IWP from $\tau$ and $r_e$, where the last two properties are derived by solar reflectances using physics-based methods. The CERES algorithms have a nighttime retrieval algorithm for these parameters, but according to Minnis et al. (2011, p. 4 386) its $\tau$ and $r_e$ values should be considered experimental.





Machine learning (ML) methods, and in particular artificial neural network (NN) approaches, are promising candidates for remote sensing retrievals. This is primarily due to that they do not require explicit assumptions used in pure physics-based models. ML methods instead can find non-linear relationships by learning from data, whether these data consist of physical observations or are obtained through physical simulations. Only concerning ice optical thickness, Yost et al. (2021) remark that

new editions of the CERES algorithms must consider NNs to improve its estimates, such as the work from Kox et al. (2014) and Minnis et al. (2016b). The Cirrus Properties from SEVIRI (CiPS, Strandgren et al., 2017) directly retrieves IWP from SEVIRI using a NN trained against Cloud-Aerosol Lidar with Orthogonal Polarization (CALIOP) observations. The CALIOP lidar signal quickly attenuates in thick clouds. Therefore, thick clouds and large IWPs are not well represented by CALIOP observations, which constrains CiPS to thin ice clouds. Holl et al. (2014), Islam and Srivastava (2015) and Mastro et al. (2022)

also directly retrieve IWP with NNs, but they make use of combinations of microwave and infrared observations, as they found this advantageous compared to just using infrared data.

In any case, the use of NNs for IWP retrieval from VISIR passive imagers remains largely unexplored. This work contributes to fill this gap for the SEVIRI instrument. Neural networks with low-latitude, Meteosat-9 SEVIRI observations are trained against DARDAR collocations. This choice of reference data for the NNs does not constrain the retrieval to small IWP

values, but rather targets the full IWP range. Retrievals with only thermal infrared (IR) channels are analyzed to overcome the daylight-only limitation of VISIR retrievals. Moreover, IR retrievals with a selection of channels based on the previous Meteosat generation are also evaluated. Additionally, the retrievals obtained here are compared with retrievals from CLAAS edition 2.1 (Finkensieper et al., 2020), a dataset based on the CPP algorithm and SEVIRI observations.

The NN method used here, quantile regression neural networks (QRNNs), was analyzed by Pfreundschuh et al. (2018)

in the context of remote sensing retrievals. QRNNs estimate the posterior distribution of Bayesian retrievals, and thus can provide uncertainties for individual retrievals. QRNN is a flexible method that allows using different NN architectures: a convolutional neural network (CNN) architecture is integrated in a QRNN to evaluate whether using of spatial information from the observations is advantageous. Both providing case-specific ML errors and using multiple footprints in the ML retrieval are new features for IWP retrievals.

IWP retrievals are the primary focus of this work. Nonetheless, two other properties are also retrieved from SEVIRI observations: the mean ice mass height and the mean ice mass size. These two variables, planned to be retrieved by the ICI product released right after its commissioning (Eriksson et al., 2020), are referred here as auxiliary variables.

## 2   Data

### 2.1   Reference data: DARDAR

The DARDAR-cloud product (Delanoë and Hogan, 2010) synergistically combines radar and lidar measurements from the CloudSat and CALIPSO tandem to provide cloud properties at a horizontal and vertical resolution of $1.4$ km and $60$ m, respectively. The CloudSat satellite mission was designed to cross the equator in ascending orbit after 13:30 local mean time, with a repeat cycle of 16 days (Stephens et al., 2002). For a given location, CloudSat-derived products, such as DARDAR, can





then only be provided at two different times, corresponding to the observations in the ascending and descending orbits, with a large time span in between. We refer to this as daytime and nighttime observations. In April 2011 CloudSat was forced to switch to daylight-only operations due to a battery anomaly, thus only allowing daytime observations (Nayak et al., 2012).

Ice water path (IWP), mean mass height ($Z_\mathrm{m}$) and mean mass size ($D_\mathrm{m}$) for an atmospheric ice column can be derived from the DARDAR cloud properties. In discrete form, these quantities are defined as

$$\mathrm{IWP} = \sum_{i \in \mathcal{Z}} \mathrm{IWC}_i \Delta z_i \tag{1}$$

$$Z_\mathrm{m} = \frac{\sum_{i \in \mathcal{Z}} z_i \mathrm{IWC}_i \Delta z_i}{\mathrm{IWP}} \tag{2}$$

$$D_\mathrm{m} = \frac{4}{(\pi \rho_\mathrm{w})^{1/4}} \frac{\sum_{i \in \mathcal{Z}} \left( \mathrm{IWC}_i^5 / N_{0,i}^* \right)^{1/4}}{\sum_{i \in \mathcal{Z}} \mathrm{IWC}_i} \tag{3}$$

where $Z_\mathrm{m}$ and $D_\mathrm{m}$ are only defined for $\mathrm{IWP} > 0$, $\mathcal{Z}$ is the set of indices defining the variable values at each DARDAR bin height, $\mathrm{IWC}_i$ ice water content, $\Delta z_i$ bin height range, $N_{0,i}^*$ the intercept parameter of the normalised size distribution of ice particles (Delanoë et al., 2005, 2014), all at bin height $z_i$, $i \in \mathcal{Z}$, and $\rho_\mathrm{w} = 1\,000\ \mathrm{kg\,m^{-3}}$ the density of water. In this work, $\mathcal{Z}$ consisted of all indices for heights above sea level. A comprehensive derivation of these variables is provided in Appendix A.

## 2.2 Input data: SEVIRI from Meteosat-9

Meteosat-9 carries the SEVIRI instrument (Aminou et al., 1997; Schmid, 2000), the MSG imager. SEVIRI allows observing the Earth in 12 spectral channels (Table 1) with a maximum repeat cycle of 15 minutes for the full Earth disc scan. The images have a sampling distance of 3 km at sub-satellite point for all channels except the high resolution visible (HRV) channel, which is 1 km. That is, the channels provide a ground resolution of $3 \times 3\ \mathrm{km}^2$ at nadir, with this resolution becoming worse when increasing the incidence angle. Therefore, SEVIRI offers a better temporal resolution and spatial coverage than DARDAR, although at a worse, varying ground resolution.

Launched in December 2005, Meteosat-9 was the primary operational satellite located at a nominal longitude of $0°$ between April 2007 and January 2013 (WMO, 2022). It has also been located at the commissioning longitude of $-6.5°$, and operational longitudes of $9.5°$, $3.5°$, and, currently, $45.5°$ (EUMESTAT, 2022). These changes in longitude make the ground resolution have a dependence on time; for a given position on Earth, only observations taken from the same operational longitude are strictly directly comparable.

The SEVIRI images are provided in a geostationary projection specified by Wolf (1999), which we refer to as SEVIRI projection. Satpy (Raspaud et al., 2021) was used to read the SEVIRI images, which retains the native SEVIRI observation grid and projection. In addition, this library automatically handles the erroneous georeferencing offset present in Meteosat images until 2017 (EUMETSAT, 2017, Sect. 3.1.4).


**Table 1.** The SEVIRI channels specification.

| Channel number | 1 | 2 | 3 | 4 | 5 | 6 | 7 | 8 | 9 | 10 | 11 | HRV |
|---|---|---|---|---|---|---|---|---|---|---|---|---|
| Nominal wavelength μm | 0.635 | 0.81 | 1.64 | 3.92 | 6.25 | 7.35 | 8.70 | 9.66 | 10.80 | 12.00 | 13.40 | 0.75 |
| Lower bound μm | 0.56 | 0.74 | 1.50 | 3.48 | 5.35 | 6.85 | 8.30 | 9.38 | 9.80 | 11.00 | 12.40 | 0.6 |
| Upper bound μm | 0.71 | 0.88 | 1.78 | 4.36 | 7.15 | 7.85 | 9.10 | 9.94 | 11.80 | 13.00 | 14.40 | 0.9 |

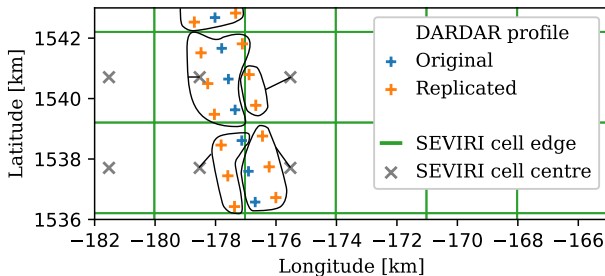

**Figure 1.** Illustration of the spatial resampling performed. The projection used is the SEVIRI projection. The DARDAR profiles are replicated from the profile in the centre of the track to the swath edges 700 m on each side. The curves enclose the profiles used to derive the averaged values in each cell.

## 2.3 Collocations

Rectified level 1.5 Meteosat-9 SEVIRI data (EUMETSAT, 2017) and DARDAR-cloud version 2.1.1 between 6 May 2008 and 31 March 2011 were collocated to form the dataset used in this work. All but a few, for practical reasons, publicly available
samples in this time range were used.

A DARDAR profile taken during a SEVIRI scan was collocated with this scan, and profiles taken in between consecutive scans were assigned the closest scan in time.

Temporally collocated DARDAR profiles were duplicated at the edges of the horizontal swath, calculated on the SEVIRI projection, to account for the DARDAR horizontal resolution. IWP, $Z_\mathrm{m}$ and $D_\mathrm{m}$ were computed for each DARDAR profile. All
variable values in a SEVIRI pixel were averaged, weighted by the profile IWP, to obtain one DARDAR value per SEVIRI pixel (Fig. 1). It can be seen that averaging all profiles and then computing IWP, $Z_\mathrm{m}$, and $D_\mathrm{m}$ from an averaged DARDAR profile is equivalent to the IWP-weighted average (see Appendix A).

Finally, the collocated images were divided in non-overlapping samples of $32 \times 32$ pixels with the DARDAR swath in the centre. The division grid of the samples was randomly placed to diminish any possible bias. Figure 2 shows the region of
interest (ROI), which ranges $[-17°, +40°]$ in longitude and $[-17°, +15°]$ in latitude. All samples covering any part of the ROI were randomly split in a training, validation and test sets of sizes 60%, 20%, and 20%, respectively, totalling more than $10^6$ pixels with reference data.



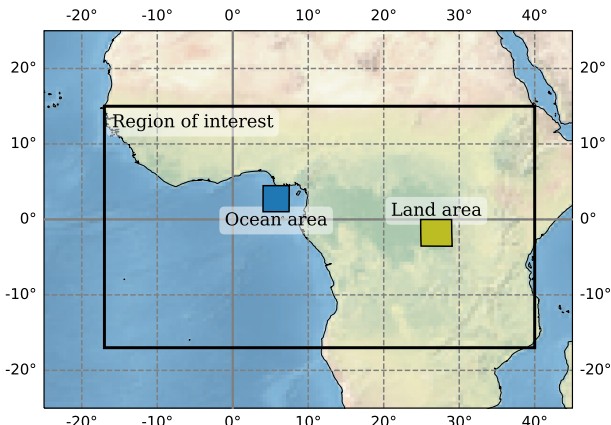

**Figure 2.** Region of interest used, and ocean and land areas used in the diurnal cycles comparison (Sect. 5.2). Ocean (land) area delimited by $[1°, 4.5°]$ ($[-3.55°, 0°]$) in latitude and $[4°, 7.5°]$ ($[24.95°, 29°]$) in longitude.

## 3 Machine learning

### 3.1 Quantile regression neural networks

For a cumulative distribution function $F_{x|\mathbf{y}}(x)$, the quantile $x_\tau$ at level $\tau \in [0, 1]$ is the value such that

$$x_\tau = \inf\{x : F_{x|\mathbf{y}}(x) \geq \tau\}. \tag{4}$$

The expectation with respect to $x$ of the loss function

$$\mathcal{L}_\tau(\hat{x}_\tau, x) = \begin{cases} \tau|x - \hat{x}_\tau| & \text{if } \hat{x}_\tau < x \\ (1 - \tau)|x - \hat{x}_\tau| & \text{otherwise} \end{cases} \tag{5}$$

is minimized by the quantile $x_\tau$ (Koenker, 2005, pp. 5–6). A quantile regression neural network (QRNN) is an artificial neural
network (NN) that seeks to minimize $\mathcal{L}_\tau$. In this work, QRNNs are employed in a multi-task learning setting for multiple quantile regression, minimizing

$$\mathcal{L}(x) = \frac{1}{|\mathcal{T}|} \sum_{\tau \in \mathcal{T}} \mathcal{L}_\tau(\hat{x}_\tau, x) \tag{6}$$

where $\mathcal{T} = \{0.01, 0.02, 0.03, \ldots, 0.98, 0.99\}$, that is, all percentiles, and $|\mathcal{T}|$ is the cardinality of $\mathcal{T}$. By extension, we use the term QRNN also for this multiple quantile regression.

The main advantage of QRNNs with respect to NNs that minimize the mean squared error (MSE) is that QRNNs can model aleatoric uncertainty. This type of uncertainty describes the inability of the observations $\mathbf{y}$ to fully determine $x$ due to hidden variables, therefore it cannot be reduced by increasing the amount of training data. QRNNs model this uncertainty by estimating $F_{x|\mathbf{y}}(x)$ at multiple quantile levels as illustrated in Fig. 3. This not only makes the regression robust against outliers





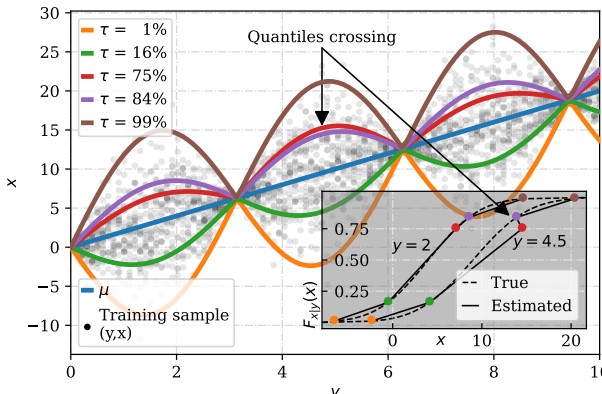

**Figure 3.** Simplified retrieval example of $x$ from $y$. A model trained to minimize the MSE can only aspire to predict the expected value of $x$ at a given $y$, indicated by the line $\mu$, while a quantile regression can describe the aleatoric uncertainty estimating quantiles at level $\tau$. These quantiles can then be used to estimate $F_{x|y}(x)$ and derive $\mu$ from it. However, quantile crossing can occur if the quantiles are not perfectly estimated.

but also provides a more complete description of the data distribution: a case-specific uncertainity can be assigned to each
prediction. Pfreundschuh et al. (2018) observed that QRNNs approximate well the posterior distribution of Bayesian remote
sensing retrievals, with uncertainty estimates consistent with non-Gaussian retrieval errors. In addition, quantile regression
enjoys the equivariance to monotone transformations property (Koenker, 2005). This allows training on a log-transformed
response variable and back-transform the estimates, a useful property for right-skewed data distributions.

Quantile crossing is the major drawback of QRNNs. This problem consists of a lack of monotonicity in the quantile estima-
tion, and is illustrated also in Fig. 3. Values derived from a QRNN experiencing severe quantile crossing can then be inaccurate.
Several approaches exist to overcome quantile crossing. In this work quantile crossing is corrected a posteriori. The correc-
tion consists of an isotonic regression of the predicted quantiles $\hat{x}_\tau$ constrained at all quantile levels. That is, the optimization
problem

$$
\begin{aligned}
\text{minimize} \quad & \sum_{\tau_i \in \mathcal{T}} (\hat{x}_{\tau_i}^{(c)} - \hat{x}_{\tau_i})^2 \\
\text{subject to} \quad & \hat{x}_{\tau_i}^{(c)} \leq \hat{x}_{\tau_j}^{(c)} \quad \forall \tau_i \leq \tau_j, \ \tau_j \in \mathcal{T}
\end{aligned}
\tag{7}
$$

is solved to find the corrected quantiles $\hat{x}_\tau^{(c)}, \tau \in \mathcal{T}$.

## 3.2 Network architectures

The QRNN approach was implemented in two different network architectures. Retrievals based on single SEVIRI pixels were
done using a multilayer perceptron (MLP). The MLP used rectified linear units as activation functions, with 16 hidden layers
with 128 hidden neurons at each hidden layer. This setup generally maximized the performance of the different retrieval
configurations.





**Table 2.** Input features used. Numbers are SEVIRI channels, and SZA satellite zenith angle.

| Settings name | VISIR | IR | IR-subset |
| --- | --- | --- | --- |
| Input features | 1–11, SZA | 5–11, SZA | 5, 9+10, SZA |

To exploit spatial correlations among neighbouring SEVIRI pixels the CNN presented in Fig. 4 was used. This CNN consists of convolutional blocks, based on the Xception network (Chollet, 2017), with an asymmetric encoder-decoder, U-net-like architecture (Ronneberger et al., 2015), and residual connections.

### 3.3 Training methodology

The $32 \times 32$ pixels images were fed to the networks using a batch size of 128 images in all trainings and network models. The input data were standardized with the training set statistics, and invalid input values replaced with $-999\,999$. The networks were trained with the Adam optimizer (Kingma and Ba, 2015) with base learning rate set to 0.001. All networks were evaluated on the validation loss as well as the number of quantile crossings, both only computed for pixels with reference values. Early stopping on the validation loss determined the selected network state. A log transform was applied to train for IWP and

$D_\mathrm{m}$. Each time the data was accessed, zero IWP values were replaced with samples from a log-uniform distribution between $10^{-8}\ \mathrm{kg\,m^{-2}}$ and $10^{-6}\ \mathrm{kg\,m^{-2}}$ (the minimum non-zero IWP in the dataset is of the order of $10^{-6}\ \mathrm{kg\,m^{-2}}$), and the images were randomly mirrored and rotated $0°$, $90°$, $180°$ or $270°$. After the first epoch, one additional pass of the training data was used to average the batch normalization statistics of each batch. These averaged statistics were then frozen and used throughout the rest of the training, empirically observed to help generalization.

The three input features settings presented in Table 2 were explored. The HRV channel was disregarded in all cases. The channel selection for the IR-subset was made to represent the Meteosat visible and infrared imager (MVIRI), the imager in the previous Meteosat generation. MVIRI only had two IR channels with spectral ranges $5.7–7.1\ \mathrm{\mu m}$ and $10.5–12.5\ \mathrm{\mu m}$; SEVIRI channel 5 covers the former range, while channel 9 and 10 cover the latter. The IR-subset had a special treatment: channel 9 and 10 inputs were averaged with weights inversely proportional to the difference between their central wavelengths and the

MVIRI central wavelength. That is, these channels were combined and fed to the network as

$$x_{\mathrm{MVIRI,11.5}} = \frac{w_9 x_9 + w_{10} x_{10}}{w_9 + w_{10}} \tag{8}$$

$$w_9 = \left(|11.5 - 10.8|\right)^{-1} = 0.7^{-1}$$

$$w_{10} = \left(|11.5 - 12.0|\right)^{-1} = 0.5^{-1}$$

where $x_9$ and $x_{10}$ are the standardized channel 9 and 10 values. This aimed to synthesise the MVIRI channel with central

wavelength $11.5\ \mathrm{\mu m}$. The satellite zenith angle was included in all three settings to take into account the varying ground resolution.

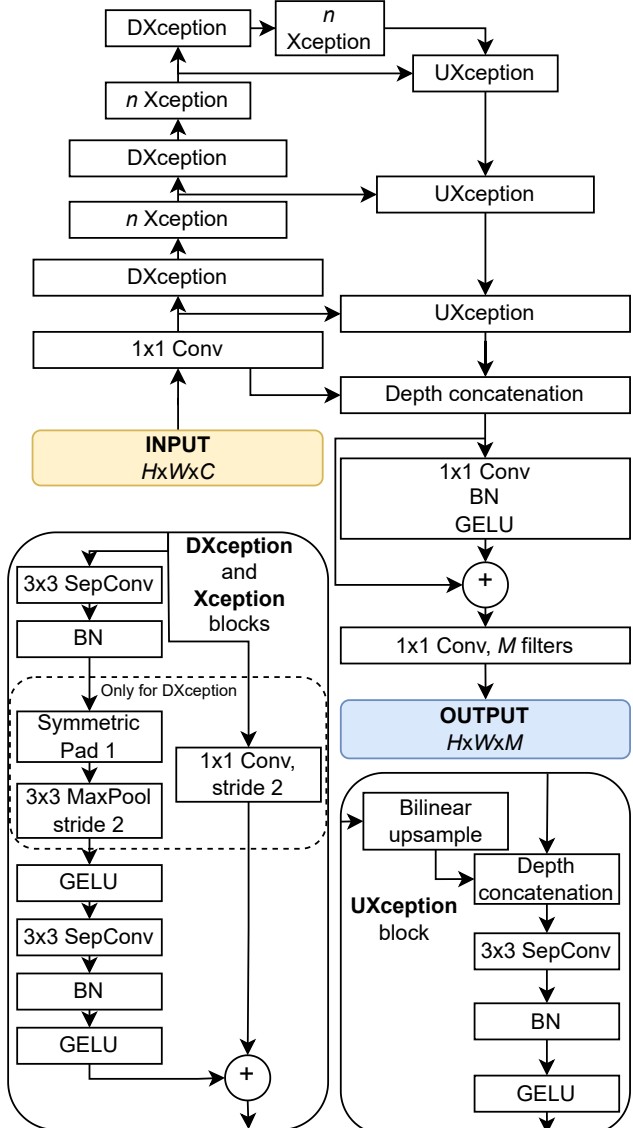

**Figure 4.** The CNN architecture used, for an input image of spatial size $H \times W$ pixels and $C$ channels, producing an output of the same dimensions with $M$ channels. Here $M = 99$, corresponding to all percentiles. Block widths relate to the spatial sizes at each stage (not to scale). 128 filters were used in all convolutional layers, and $n$ Xception means that $n$ consecutive Xception blocks are applied, where it was chosen $n = 2$. Depthwise separable convolutions (SepConv), with a $3 \times 3$ kernel, preserve the spatial size using a replicate padding of 1 at the depthwise convolution. GELU: Gaussian Error Linear Unit (Hendrycks and Gimpel, 2020), BN: Batch Normalization (Ioffe and Szegedy, 2015), $1 \times 1$ Conv: pointwise convolution. Strides of 1, otherwise indicated.





## 4 Retrieval results

We examined two aspects for retrieving IWP from Meteosat-9 SEVIRI images: the channel selection and the use of spatial information. The evaluation of QRNNs or, more generally, probabilistic predictions is not straightforward. The common summary statistics root mean squared error (RMSE), mean absolute error (MAE), and bias require a point estimate. QRNNs do not provide a unique point estimate, therefore one value has to be selected to compute these statistics.

Throughout this section, we use the expected value (mean) of the distribution as the QRNN point estimate. The distribution was obtained by constructing a cumulative distribution function from linearly interpolating the predicted quantiles, as illustrated in Fig. 3, and linearly extrapolating quantiles at level $\tau \in \{0, 1\}$ from the two nearest quantiles. That is, a continuous distribution function was constructed from each QRNN prediction. Quantiles at level $\tau = 0$ were clipped to zero to avoid implausible negative values resulting from the linear extrapolation.

A probabilistic measure of performance is the continuous ranked probability score (CRPS), which here is defined for each prediction as

$$\text{CRPS} = \int\limits_{-\infty}^{+\infty} \left[ \hat{F}_{x'|\mathbf{y}}(x') - \mathbb{1}\left(x' \geq x\right) \right]^2 dx \tag{9}$$

where $\mathbb{1}$ is the indicator function, $x$ is the scalar reference value, and $\hat{F}_{x'|\mathbf{y}}(x')$ is the cumulative distribution function estimated with the QRNN. We denote the mean and median of all CRPS values from a QRNN as $\text{CRPS}_\mu$ and $\text{CRPS}_m$, respectively.

The RMSE, MAE, and bias can be relatively misleading if the data ranges several orders of magnitude, as in the case of IWP. Therefore, the QRNNs should not be judged only on these summary statistics but rather mainly with the plots presented in, for example, Fig. 5. Furthermore, it should not be concerning that quantiles at extremal quantile levels, such as levels $\tau \in \{0.01, 0.02, 0.98, 0.99\}$, show unrealistic values. These quantiles have a small contribution and we observed that they can show noticeable variations between different trainings.

Another summary statistic can be computed from point estimates: the correlation between the retrieved value and the reference value. The Spearman correlation coefficient $r_S$ is used here. This statistic measures the monotonic relationship between two variables, where $r_S = \pm 1$ imply perfect correlation, and 0 no correlation at all.

Finally, the shape of a predicted distribution by a QRNN can be analyzed computing its skewness $\gamma_1$ and kurtosis $\beta_2$. For a random variable $X$, they are defined by

$$\gamma_1 = \mu_3 / \mu_2^{3/2} \tag{10}$$
$$\beta_2 = \mu_4 / \mu_2^2 \tag{11}$$
$$\mu_t = \mathbb{E}[(X - \mathbb{E}[X])^t]. \tag{12}$$

The skewness measures the distribution asymmetry about its mean: the more positive $\gamma_1$ is, the larger the right-skew, and the more negative $\gamma_1$, the larger the left-skew. On the other hand, kurtosis measures the contribution of the tails to the rest of the distribution.

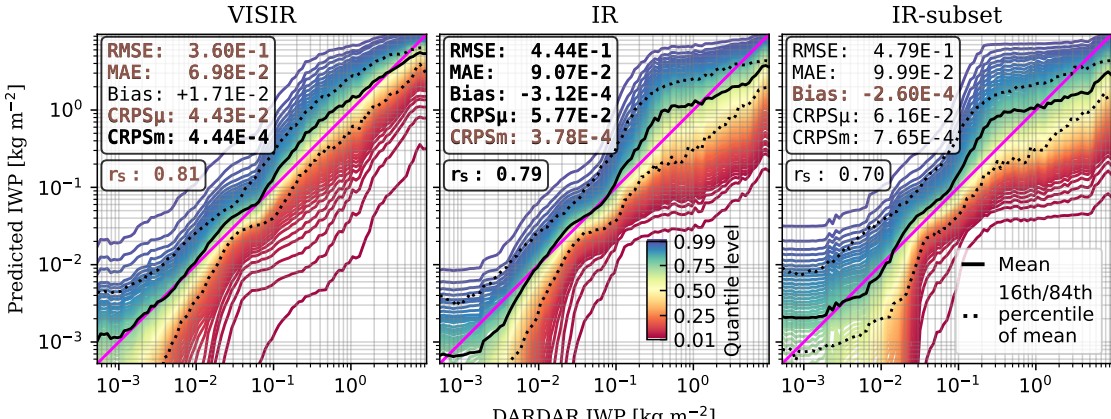

**Figure 5.** IWP predictions with MLP networks for the different SEVIRI channels as input (Table 2). Networks trained only with daytime observations. The solid curves indicate the median value of each prediction at the local DARDAR IWP. The statistics are computed for the expected value. In brown the best value, and in bold font the second best value. Summary statistics in $\mathrm{kg\,m^{-2}}$. Statistics and predictions computed using all test data.

## 4.1 Channel selection

We used MLPs to analyze the selection of input features. The trainings performed for this analysis only used daytime samples
to facilitate the networks to leverage the visible channels. Figure 5 summarises the findings for the test set.

We make two main observations. Firstly, the predicted distributions are right-skewed, as the expected value is larger than the median ($\tau = 0.5$). Figure 6 shows skewness and kurtosis frequencies for MLP predictions. It is observed that the retrieval distributions tend to be non-Gaussian. As a reference, Gaussian distributions have $\gamma_1 = 0$ and $\beta_2 = 3$. It can also be observed that VISIR retrievals tend to be more skewed and have more information in the tails than the two other options. Secondly, the
visible channels are useful for the retrieval of larger IWP values. This is not only observed from the expected value being closer to the identity line in the range $10^{-1}$–$10^{+1}$ $\mathrm{kg\,m^{-2}}$, but also from that the model is more confident as there is less spread in the predicted quantiles, particularly for larger IWP values.

Although using the VISIR setting favours the retrieval, this setting is restricted to daytime retrievals. This implies that the retrieval performance has a dependence on the solar angles. Because of the CloudSat orbit (Sect. 2), there is little variation in
the solar angles range in the DARDAR data used. It is erroneous to execute QRNN VISIR retrievals from observations with solar angles not found in the training data. An IR-only retrieval is thus preferred for a constant performance throughout the full day, as it is independent of the solar angles.

Concerning the IR and IR-subset settings, the main difference between the retrievals resides in the confidence of the models. Particularly at low IWP, there is more spread among the quantiles for the IR-subset retrieval. All summary statistics except bias
also disfavour the IR-subset setting, although it can be argued the two options have a similar performance.



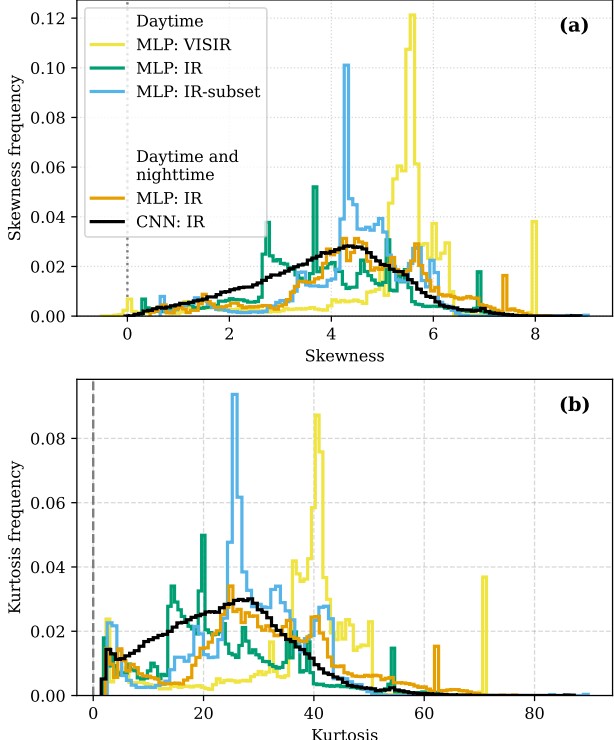

**Figure 6.** Skewness (a) and kurtosis (b) frequencies for all QRNN predictions with DARDAR IWP between $10^{-3}\,\mathrm{kg\,m^{-2}}$ and $10^{+1}\,\mathrm{kg\,m^{-2}}$, indicating the network architecture, input settings and training observations used (corresponds to results shown in Figs. 5 and 7). Same legend for (a) and (b).

## 4.2 Spatial information

Building on the previous section arguments for IR-only retrievals, we examined the use of spatial information for the IR input setting. Figure 7 shows the retrieval performance with and without the use of spatial information, corresponding to the CNN and MLP, respectively. In this case, both daytime and nighttime observations were used for training and evaluation of the models. It can be observed that using spatial information improves the retrieval of IWP: not only it tends to be closer to the identity line, but also shows better summary statistics, except for the bias.

The main advantage of using the CNN is a decrease in uncertainty. This argument comes from a lower CRPS for the CNN, as well as smaller spread for the expected value with respect to the reference values. That is, the CNN network likely leverages local spatial patterns to improve the retrieval uncertainty. Analogously to Sect. 4.1 and based on Fig. 6, it can also be inferred that retrieval distributions constructed from the predicted quantiles are not necessarily Gaussian. In this case, however, the CNN distributions tend to be less skewed and have lighter tails than the MLP, which can be considered preferable.



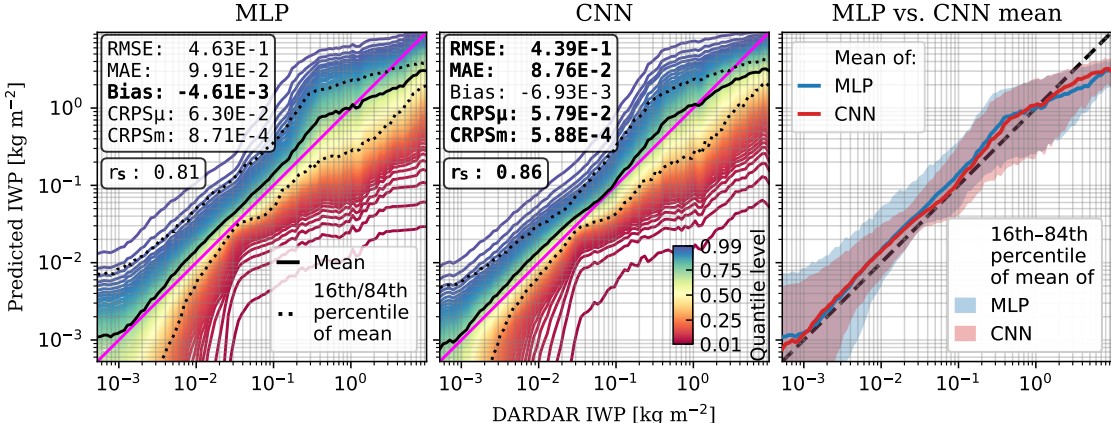

**Figure 7.** IWP predictions using spatial information (CNN) and without using spatial information (MLP). Networks trained with both daytime and nighttime observations with the IR input settings (Table 2). The solid curves indicate the median value of each prediction at the local DARDAR IWP. The statistics are computed for the expected value, and the bold font indicates the best value. Summary statistics in $\mathrm{kg\,m^{-2}}$. Statistics and predictions computed using all test data. The rightmost plot corresponds to the black curves from the other plots.

### 4.3 Auxiliary variables

Given that the CNN showed to reduce the retrieval uncertainty, we analysed the retrieval of $Z_\mathrm{m}$ and $D_\mathrm{m}$ with the same CNN architecture. The data employed for training and evaluating the networks was the same as in Sect. 4.2. Nevertheless, the networks struggled to model $D_\mathrm{m}$ when this variable was derived from columns with low IWP. Excluding all $D_\mathrm{m}$ with IWP $\leq$ $10^{-3}\ \mathrm{kg\,m^{-2}}$ from the dataset enabled the networks to model it. This situation was not experienced when training for $Z_\mathrm{m}$, but for simplicity $Z_\mathrm{m}$ values with IWP $\leq 10^{-3}\ \mathrm{kg\,m^{-2}}$ were also excluded.

The retrievals of $Z_\mathrm{m}$ and $D_\mathrm{m}$ as well as the distributions of the training set are presented in Fig. 8. The expected value of $Z_\mathrm{m}$ (Fig. 8a) follows closely the identity line for 10–15 km, and the expected value of $D_\mathrm{m}$ (Fig. 8b) for 0–200 μm, in both cases with relatively low spread. Nevertheless, comparing them with the probability distribution functions (PDFs), it should not be surprising, as the models can leverage a priori information for the retrieval in these cases. Furthermore, any effects of multilayer clouds in the $Z_\mathrm{m}$ retrieval are unclear, as well as the relationship between retrievals of $Z_\mathrm{m}$ and cloud top heights, questions which can be considered for further research.

### 5 Comparison with CLAAS

The CLAAS dataset edition 2.1 (Finkensieper et al., 2020) provides cloud properties derived from MSG satellites. One of the cloud properties provided in this dataset is IWP. The retrieval in CLAAS is based on the two-stage CPP algorithm (CM SAF, 2016). In the first stage of the algorithm the cloud type is determined using an IR-based algorithm. The cloud type is then reduced to a cloud top phase indicator: clear, liquid or ice. In the second stage, the cloud optical thickness $\tau$ and the particle



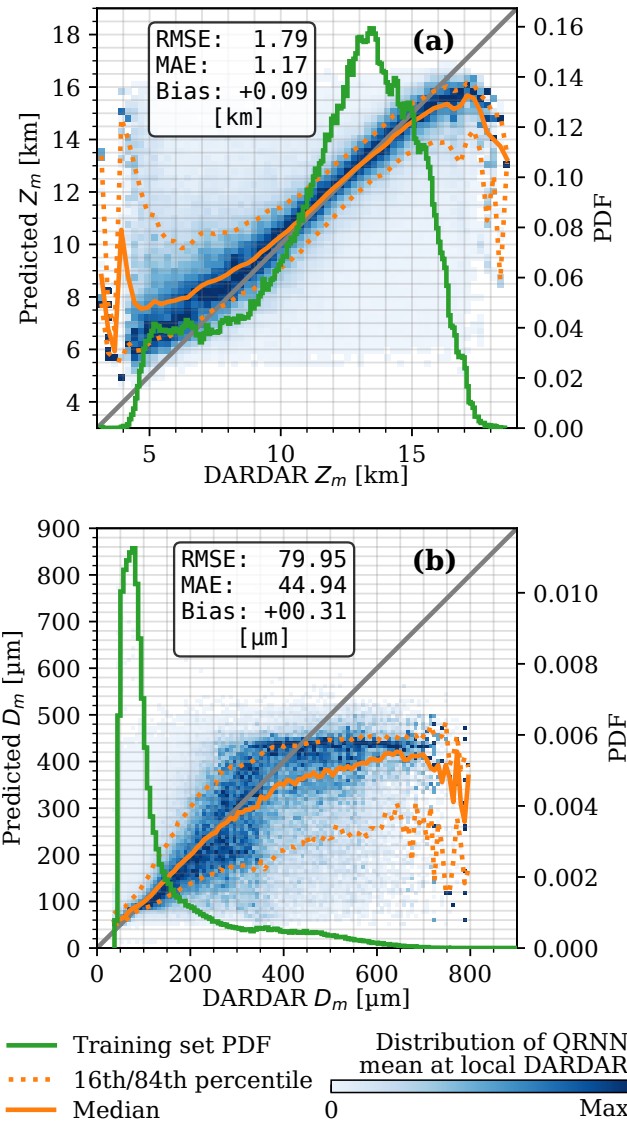

**Figure 8.** Predictions of $Z_m$ (a) and $D_m$ (b) with the CNN for the test set, and training set PDF. Networks trained with both daytime and nighttime observations. Statistics plotted by the orange curves derived from the distributions indicated by the colour bar.

effective radius $r_e$ are computed for cloudy pixels. This retrieval uses the SEVIRI channels 1 and 3, and compares the observed

reflectances to look-up tables of simulated reflectances. Following Stephens (1978), IWP in CLAAS is calculated as

$$IWP = 2\tau r_e \rho / 3 \tag{13}$$

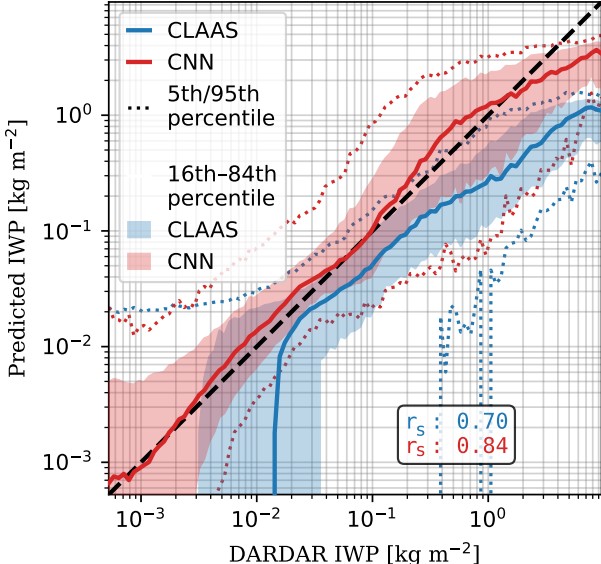

**Figure 9.** IWP predictions from our approach (QRNN with CNN architecture, IR input settings) and from CLAAS. The QRNN is trained with daytime and nighttime observations, but the test data in both cases contains only daytime observations. The QRNN expected value is used as its retrieval estimate. The solid curves indicate the median value of each measure at the local DARDAR IWP. Summary statistics in $\mathrm{kg\,m^{-2}}$. Predictions and statistics computed for all test set samples also available in CLAAS.

using $\rho = 930 \ \mathrm{kg\,m^{-3}}$ for the ice density. An inconvenience with the CPP algorithm is the variability of $r_e$ throughout thick ice clouds. This may make $r_e$ totally unrepresentative of the ice column in Eq. (13). A further inconvenience is the channel selection, which only allows retrieving IWP for daytime observations, and can be affected by sunglints.

CLAAS has been thoroughly validated (Benas et al., 2017; CM SAF, 2020b) and, despite its limitations for IWP retrieval, it can be considered a reasonable dataset of reference. We compared our CNN, IR-only, QRNN IWP retrieval with two products in CLAAS: the IWP instantaneous retrieval and its monthly mean diurnal cycle.

## 5.1    Instantaneous IWP retrieval

The instantaneous IWP product from CLAAS corresponds to retrieving IWP from each Meteosat observation with the CPP
algorithm. For the observations in our test set, all matching CLAAS retrievals used Meteosat-9 observations. All daytime observations in the test set also present in CLAAS were used in this comparison.

The CLAAS-2.1 instantaneous IWP is provided at the SEVIRI native grid, and it also suffers from the erroneous georeferencing offset in Meteosat images (CM SAF, 2020a, pp. 14–15). The offset in the CLAAS data was also handled with Satpy (Raspaud et al., 2021). DARDAR data was collocated with CLAAS following the method described in Sect. 2.3, assuming
zero IWP for the CLAAS data when the cloud phase indicated by CLAAS is not ice.





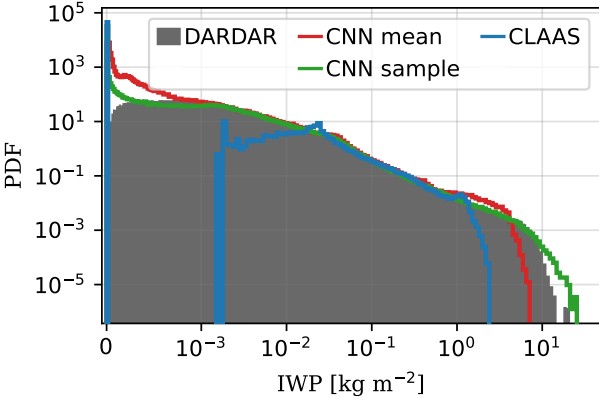

**Figure 10.** PDFs from the different IWP retrievals in Fig. 9 (CNN mean, CLAAS, and DARDAR curves), estimated with a histogram. CNN sample indicates the QRNN retrieval where one QRNN sample replaces the mean as the point estimate. Linear scale in the horizontal axis between 0 and $10^{-3}$ kg m$^{-2}$ and log scale afterwards.

Figure 9 compares the retrieval in CLAAS and the QRNN approach. It is seen that the QRNN mean has a tendency to be closer to DARDAR. This should not be surprising given that the network learns from DARDAR data. Furthermore, the retrieved $r_e$ with the CPP algorithm is likely smaller than the DARDAR retrieval (CM SAF, 2020b), and therefore the estimated IWP with Eq. (13) should be smaller. The CPP algorithm also characterizes a cloud as either of water or ice phase, ignoring mixed-295 phase states. This restriction is not done in the QRNN retrieval. This dichotomy in the CPP algorithm might be related with the abrupt drop of its curve in Fig. 9. Besides this, the QRNN retrieval has a better monotonic relationship with DARDAR, indicated by the higher $r_S$.

The CLAAS dataset evaluation from Benas et al. (2017) provides an analogous Fig. 9, which is different from the results presented here. We note, however, their different collocation strategy, that they do not restrict the collocations to the region of 300 interest of this work, and, significantly, that they exclude any DARDAR profiles that are not only of ice cloud phase.

The IWP probability distribution functions of the retrievals under comparison are shown in Fig. 10. It is observed that the expected value of the QRNN can retrieve larger values of IWP than the CPP algorithm. The difference of the CNN PDF with respect to the DARDAR PDF might be corrected a posteriori, and can be considered for further research. Figure 10 also presents the case when one sample from the QRNN is used as a point estimate instead of the mean. The PDF constructed 305 from this point estimate follows closer the DARDAR PDF, and shows that the QRNN uncertainty can capture DARDAR IWP values well from IR-only observations, even the larger values. The ability to capture larger IWP values combined with a better correlation with DARDAR show that the CNN, IR-only QRNN retrieval performs better than the CPP algorithm retrieval for IWP retrievals.



## 5.2 Monthly mean diurnal cycles

The monthly mean diurnal cycles product in CLAAS is obtained by averaging hour-wise all observations over all days of a month (CM SAF, 2020a). The files for this CLAAS product covering the full 2012 year indicate that Meteosat-9 was the data source. Consequently, and because the training did not contain data for 2012, we replicated this product for the CNN, IR input settings QRNN retrieval using all Meteosat-9 observations taken in 2012, except for one clearly faulty observation.

Two tropical areas were used for comparing the monthly mean diurnal cycles, denominated as ocean and land areas (Fig. 2).
These areas were chosen based on Fig. 10e from Benas et al. (2017), where it is seen that they have high IWP on average. All pixel values in them were averaged to compute a single monthly mean diurnal cycle per area, and the results are shown in Fig. 11.

There are two primary differences between the diurnal cycles from the two methods. There is a general discrepancy in the IWP magnitude, with this being lower in CLAAS. This agrees well with the remarks in the previous section, where CLAAS
IWP results smaller than the QRNN retrieval, as well as than DARDAR IWP. Mean IWP values significantly lower than DARDAR ones are also observed in similar retrievals, such as retrievals from MODIS (Duncan and Eriksson, 2018). Secondly, CLAAS does not retrieve IWP during nighttime, an inconvenience that our IR-only retrieval does not present.

Figure 11 also shows the local solar time (LST) coverage of all samples in the training set. It is seen that the network learnt to make use of the physical information from only roughly 25 minutes of LST in either daytime or nighttime to make retrievals
at other times. This was possible as a consequence of selecting the QRNN that only uses IR channels.

It is worth noting that an exhaustive validation of the diurnal cycles is impossible as there is no DARDAR data outside these time ranges for both areas. In addition, currently available DARDAR version 2.1.1 data for 2012 is scarce and only for daytime: the number of DARDAR retrievals in a month ranges from zero to six overpasses for the chosen areas. This makes the spatial and temporal coverage excessively sparse to estimate a comparable monthly mean value from DARDAR. Bearing this in mind,
and that IR retrievals do not depend on solar angles, it is reasonable again to assume that the QRNN retrievals using the CNN and IR input settings are more accurate than the physics-based CPP algorithm.

## 6    Discussion

The performance of any ML retrieval will depend on, among other factors, its capacity to learn from the training data, its expressivity to represent the retrieval complexity, and its ability to generalize to unforeseen data. In the case of NNs, these
factors can be tackled by exploring different architectures and training methodologies. Achieving better ML results depends on the computational resources available. However, the quality of the training data is a main determinant.

The results presented here show that QRNNs learn to represent DARDAR retrievals from SEVIRI observations to a certain extent, where DARDAR is considered a ground truth. However, DARDAR contains retrieval errors; it does not necessarily represent the exact atmospheric state. In DARDAR-cloud version 3, the retrievals show a 24% reduction in IWP on average
(Cazenave et al., 2019). Nonetheless, this work used an older version to compare with the validation works of the CLAAS



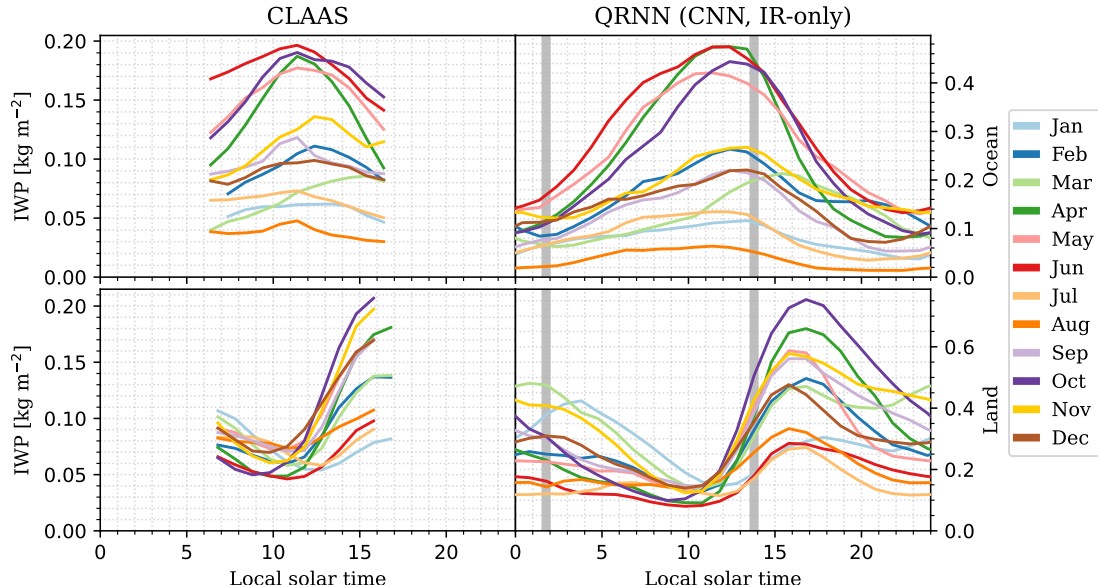

**Figure 11.** Monthly mean diurnal cycles for 2012 from CLAAS and the CNN, IR-only QRNN retrieval for the areas in Fig. 2. The QRNN expected value is used as the retrieval value. Local solar time (LST) approximated from UTC as $\mathrm{LST} = \mathrm{UTC} + 12\,\mathrm{hours}/180° \cdot \mathrm{longitude}$. The grey areas indicate the LST coverage of the DARDAR profiles in the training set. Note the different vertical axes.

dataset (Benas et al., 2017; CM SAF, 2020b). If a product is to be created from this work, then a subsequent refined DARDAR product should be used for the best quality of the reference data.

Concerning NN architectures, the CNN improved the IWP retrieval performance by decreasing the QRNN uncertainty. Moreover, it generally presents better summary statistics than its MLP counterpart, except for the bias. There is no reason to
believe that a CNN approach inherently leads to a worse bias. Training the selected CNN is hard: nearly twice more parameters are optimized in the CNN than in the MLP ($532\,787$ and $304\,169$ parameters, respectively). Therefore, better trainings or other CNNs can reduce the bias.

The retrievals for an image with the CNN can appear smoothed out when compared to the MLP retrievals. A particular example is presented in Fig. 12. This can be a consequence of using spatial information. DARDAR consists of narrow stripes,
hence there is few neighbouring SEVIRI pixels with collocated reference data: this scarcity in spatial reference information may discourage predicting large differences between neighbouring pixels.

The DARDAR narrow stripes also make it unfeasible to determine, visually, which network architecture would produce retrievals from SEVIRI IR images that differ less from a hypothetical analogous DARDAR image. The networks performance can then only be evaluated on the results presented here, on visual animations, or compared with retrievals from non-geostationary
satellites, which consequently cannot provide the same temporal coverage.





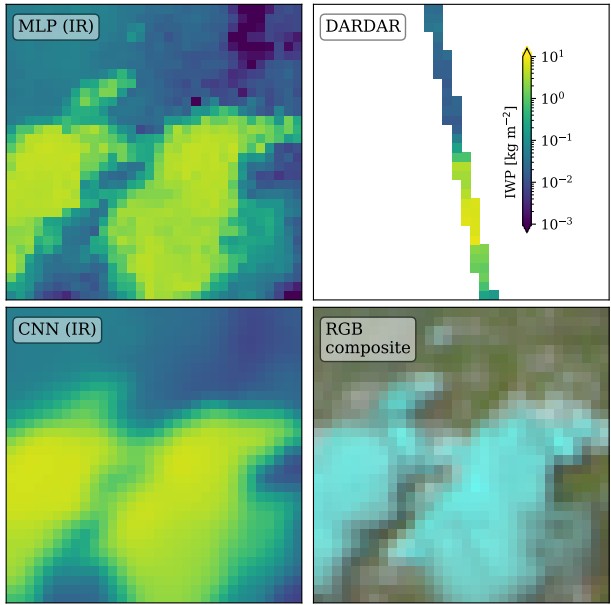

**Figure 12.** IWP retrievals (same colour scale) from a sample where the CNN may smooth out the retrieved value. The RGB composite is generated with channels 1 (blue), 2 (green), and 3 (red), using the `natural_color` composite from Satpy (Raspaud et al., 2021). Ice clouds strongly absorb in $\lambda = 1.64\,\mu m$, hence appear bluish.

Visual animations for the ocean and land areas for retrievals from all observations in January 2012 are provided as a video supplement, where Fig. 13 illustrates few of these retrievals. These animations show the IWP retrieval by the MLP and the CNN, both using only IR channels, and the QRNN expected value as the point estimate. It is observed that while both networks generally agree when there is IWP, they can show clear differences in the retrieved values: in Fig. 13a-b, the CNN retrieval

clearly differentiates between the left and right sides with respect to the MLP retrieval. Additional research might be able to explain the cause of such differences. On the other hand, it can be considered that the CNN qualitatively favours the IWP retrieval in the time dimension, since the MLP temporal evolution for one pixel exhibits a small noise-like pattern (Fig. 13c-f show a few contiguous retrievals, but it is clear in the video supplement).

This work covered only retrievals from Meteosat-9, constrained by the range of daytime and nighttime observations from

CloudSat. Retrievals from Meteosat-9 can complement DARDAR, in both time and spatial dimensions, by providing nearly-instantaneous retrievals for positions not even sampled by CloudSat. The models developed can further complement DARDAR if they are transferred to other MSG satellites. This would allow covering a much larger time span of IWP retrievals. This model transferability should, in principle, be possible as all carry the SEVIRI instrument, and can be a line of research.

A more demanding challenge would be that of transferring the models to the previous or next Meteosat generation for an

even larger coverage: approaches such as the one performed here, where a subset of IR channels were used to approximate the previous Meteosat imager (Sect. 3.3), may be too simplistic to represent data from the actual instruments. Whether the



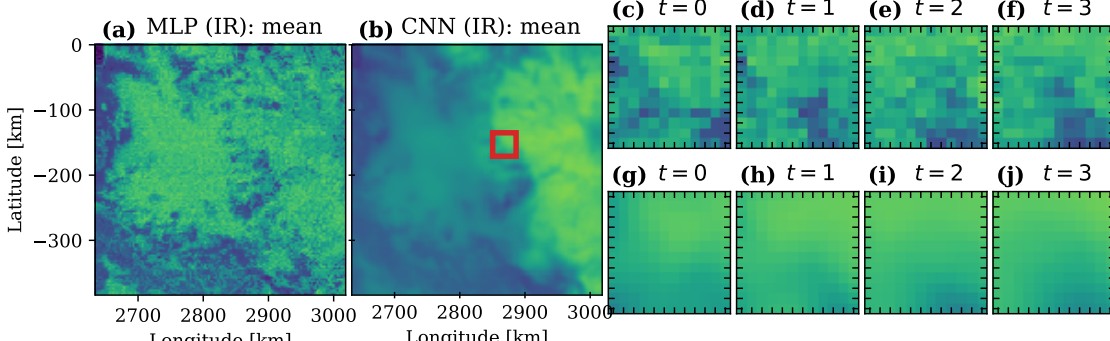

**Figure 13.** In (a-b), one frame from the visual animations showing IWP retrievals from the IR input setting corresponding to the first observation of 2012 (nighttime) over the land area. MLP retrievals in (a), where the mean is used as a point estimate, and analogously for CNN retrievals in (b). Panels (c-f) show the retrieval at $t$ observations from the retrieval in (a), zoomed into the area shown in red in (b); analogously for (g-j) for (b). Same colour scale as in Fig. 12.

models are transferred to other MSG satellites or other Meteosat generations, or even for the shifts in the Meteosat-9 nominal longitude, a comprehensive validation against DARDAR is impossible due to the time period it has covered.

SEVIRI is an instrument that was not specifically designed to characterize atmospheric ice. Comparing with the expected
retrieval performance from ICI (Eriksson et al., 2020, Fig. 9a), the IWP retrievals obtained here are not too distant from the upcoming ICI retrievals, where the latter follows closely the identity line for IWP $> 3 \times 10^{-2} \, \text{kg m}^{-2}$, but the IR-only, QRNN IWP retrievals from SEVIRI are better for IWP $< 10^{-2} \, \text{kg m}^{-2}$. However, the IWP retrievals from ICI will, in principle, have much less uncertainty. The same can be said for $Z_\text{m}$. Nevertheless, ICI is expected to be able to retrieve $D_\text{m}$ much better: the ICI retrieval follows closely the identity line, even for the larger $D_\text{m}$ values, and it also presents less uncertainty. Overall, ICI
will provide more accurate retrievals.

Nonetheless, the sun-synchronous orbit of Metop-SG B, the satellites that will carry ICI, poses a similar challenge for temporal coverage as CloudSat, although the ICI wide swath will provide semi-global coverage on a daily basis. If the models developed here are transferred to other current or upcoming Meteosat satellites, then near-instantaneous retrievals from a Meteosat satellite can complement the more accurate ICI retrievals. In addition, the finer spatial resolution of SEVIRI can
further complement the coarser ICI resolution. A further consideration is that there will be no collocations between ICI and space-based cloud radars (CloudSat and EarthCARE) at lower latitudes as they are all in sun-synchronous orbits but will cross the equator at different local times. One way to overcome the time difference in observations would be to use retrievals like the ones presented here as a common reference.

Covering the observational gaps from ICI and space-based cloud radars with geostationary data allows obtaining diurnal
cycles as in Sect. 5.2. Cloud diurnal cycles are a feature often overlooked. Concerning IWP on a regional scale, limb sounding observations have revealed discrepancies with global climate models (Eriksson et al., 2014; Jiang et al., 2015). Discrepancies of the diurnal cycle of clouds in such models increase uncertainties in climate projections (Yin and Porporato, 2017). The avail-



ability of diurnal cycles from satellite observations should help obtain more realistic model simulations by better constraining them.

The results presented in this work lead to extending the work to a larger ROI. However, the worse resolution of geostationary images when increasing the incidence angle comes at a cost. Firstly, it implies more difficulties in accurately resolving the actual atmospheric state. Secondly, the DARDAR data results of less quality when resampled to match the geostationary images. On the one hand, the larger the incidence angle, the more DARDAR profiles are located in one SEVIRI pixel. On the other hand, pixels at large incidence angles contain more aggregated information than at smaller angles. These inconveniences, all product

from the worse resolution, create a significant irregularity in the reference data. Extending this work to a full-disc retrieval can benefit from handling this quality irregularity; more advanced approaches than just using the satellite zenith angle can be considered for this issue. Further research can include using uncertainty in the reference data when training the models, where this uncertainty can be based on the disagreement among the DARDAR profiles in a SEVIRI pixel.

## 7   Conclusions

Progress in characterizing the mass of ice hydrometeors helps to model better the climate sensitivity and changes in precipitation. The frequent observations from geostationary satellites can contribute to this progress, but IWP retrievals from geostationary radiances are complex. Traditional approaches for these retrievals present limitations: among others, they rely on solar reflectances and estimate IWP indirectly. Machine learning models designed to avoid such inconveniences can be trained to approximate IWP retrievals from active sensors.

In this work, QRNNs are used to retrieve IWP from the SEVIRI instrument aboard Meteosat-9. QRNNs reproduce Bayesian retrievals, and it is seen that the IWP retrievals tend to be non-Gaussian in the different configurations explored. The use of solar reflectances helps the QRNN retrieval, but this restricts the retrieval to the small range of daytime solar angles in the reference data. Hence, thermal IR retrievals are preferred. A subset of IR channels based on the previous Meteosat generation satellites shows promising results, but further work is required to evaluate the compatibility with their imagers.

Spatial information incorporated in the QRNN through a CNN improves the IWP retrieval. Therefore, retrievals from a CNN, IR-only QRNN trained with daytime and nighttime observations are advantageous. The QRNN retrievals based on an IR-only CNN not only provide retrievals at any time of the day but also approximate better DARDAR than the physics-based CPP algorithm. Overall, these IR retrievals suggest extending the work to cover larger ROIs, as well as considering machine learning retrievals in the preparations for upcoming missions.

There will be no collocations between the upcoming ICI and space-based cloud radars at tropical latitudes. Geostationary data can then act as a bridge in time between the two types of observations. For the ROI used in this work, which consists of low latitudes, retrievals of IWP as well as $Z_m$ from geostationary radiances can complement in time and space ICI, but $D_m$ retrievals are far from the expected ICI performance. In addition, the models trained in this work also provide a basis for retrievals from other Meteosat satellites in the current and next generations.



*Code and data availability.* The code used to produce the results in this work is publicly available at https://doi.org/10.5281/zenodo.6570587 (Amell, 2022a). The code also indicates how to replicate the dataset used in this work, where the source data DARDAR-cloud version 2.1.1 is available at https://www.icare.univ-lille.fr/dardar/data-access/ (last access 8 May 2022), Meteosat-9 level 1.5 data at https://data.eumetsat.int/data/map/EO:EUM:DAT:MSG:HRSEVIRI (last access 8 May 2022), and the CLAAS-2.1 products used consisted of the instantaneous COT, CPH and CWP (CPP) product and the monthly mean diurnal-cycle product (Finkensieper et al., 2020).

*Video supplement.* IWP retrievals from all January 2012 Meteosat-9 observations for both ocean and land areas are found at https://doi.org/10.5281/zenodo.6639443 (Amell, 2022b).

## Appendix A: Derivation of IWP, $Z_\mathrm{m}$ and $D_\mathrm{m}$ from DARDAR

Let $z_0$ and $z_\mathrm{max}$ be the range of heights in which we are interested, $\mathcal{P}$ define the set of indexes identifying the profiles we want to combine, and $|\mathcal{P}|$ its cardinality. DARDAR variables are provided at a discrete range of heights. Let $\mathcal{Z}$ contain the set of indexes identifying the DARDAR heights between $z_0$ and $z_\mathrm{max}$, and $\Delta z_i$ the bin height range at height $z_i$, $i \in \mathcal{Z}$. Furthermore, let $w_l$ define the importance of profile $l \in \mathcal{P}$. Throughout these derivations, we assume that all profiles are equally important, and the subindexes refer to the DARDAR variable values at the corresponding height and profile. The variables $Z_\mathrm{m}$ and $D_\mathrm{m}$ are only defined when IWP $> 0$, that is, $\mathrm{IWC}_i \neq 0$ for some $z_i$.

Concerning the average IWP from a set of profiles we have

$$\mathrm{IWP} = \int_{z_0}^{z_\mathrm{max}} \mathrm{IWC}(z)\,dz = \frac{\sum_{l\in\mathcal{P}}\sum_{i\in\mathcal{Z}} w_l \mathrm{IWC}_{il}\Delta z_{il}}{\sum_{l\in\mathcal{P}} w_l} \tag{A1}$$

$$= \frac{1}{|\mathcal{P}|}\sum_{l\in\mathcal{P}}\sum_{i\in\mathcal{Z}} \mathrm{IWC}_{il}\Delta z_{il} \tag{A2}$$

where we see that IWP from a set of profiles is the arithmetic mean of each profile IWP.

Regarding $Z_\mathrm{m}$ we have

$$Z_\mathrm{m} = \frac{\int_{z_0}^{z_\mathrm{max}} z\mathrm{IWC}(z)\,dz}{\int_{z_0}^{z_\mathrm{max}} \mathrm{IWC}(z)\,dz} \tag{A3}$$

$$= \frac{\left(\sum_{l\in\mathcal{P}} w_l\right)^{-1}\sum_{l\in\mathcal{P}}\sum_{i\in\mathcal{Z}} w_l z_{il}\mathrm{IWC}_{il}\Delta z_{il}}{\left(\sum_{k\in\mathcal{P}} w_k\right)^{-1}\sum_{k\in\mathcal{P}} w_k\mathrm{IWP}_k} \tag{A4}$$

$$= \sum_{l\in\mathcal{P}} \frac{\mathrm{IWP}_l}{\mathrm{IWP}_l}\frac{\sum_{i\in\mathcal{Z}} z_{il}\mathrm{IWC}_{il}\Delta z_{il}}{\sum_{k\in\mathcal{P}}\mathrm{IWP}_k} = \frac{\sum_{l\in\mathcal{P}}\mathrm{IWP}_l Z_{\mathrm{m},l}}{\sum_{k\in\mathcal{P}}\mathrm{IWP}_k}. \tag{A5}$$

Therefore, the $Z_\mathrm{m}$ of a set of profiles is given by averaging the $Z_\mathrm{m}$ of each profile weighted by its IWP.

To derive the expression for $D_\mathrm{m}$ we first need to introduce the diameter of an equivalent melted particle $d_\mathrm{eq}$, the particle size distribution for diameter $d_\mathrm{eq}$ and height $z$ as $n(d_\mathrm{eq}, z)$, and $\rho_\mathrm{w} = 1\,000\,\mathrm{kg\,m^{-3}}$, the water density. From Delanoë et al. (2014),


we have that

$$\mathrm{IWC}(z) = \frac{\pi \rho_\mathrm{w}}{6} \int\limits_0^{+\infty} d_\mathrm{eq}^3 n(d_\mathrm{eq}, z)\, dd_\mathrm{eq} \tag{A6}$$

and

$$\int\limits_0^{+\infty} d_\mathrm{eq}^4 n(d_\mathrm{eq}, z)\, dd_\mathrm{eq} = \left[ \frac{4^4 \left( \int_0^{+\infty} d_\mathrm{eq}^3 n(d_\mathrm{eq}, z)\, dd_\mathrm{eq} \right)^5}{6 N_0^*(z)} \right]^{1/4} \tag{A7}$$

$$= \left[ \frac{4^4}{6} \left( \frac{\mathrm{IWC}(z)}{\pi \rho_\mathrm{w}/6} \right)^5 \frac{1}{N_0^*(z)} \right]^{1/4}. \tag{A8}$$

The $D_\mathrm{m}$ for a profile $l$ results

$$D_{\mathrm{m},l} = \frac{\int_{z_0}^{z_\mathrm{max}} \int_0^{+\infty} d_\mathrm{eq}^4 n(d_\mathrm{eq}, z)\, dd_\mathrm{eq} dz}{\int_{z_0}^{z_\mathrm{max}} \int_0^{+\infty} d_\mathrm{eq}^3 n(d_\mathrm{eq}, z)\, dd_\mathrm{eq} dz} \tag{A9}$$

$$= \frac{\sum_{i \in \mathcal{Z}} \Delta z_{il} \int_0^{+\infty} d_\mathrm{eq}^4 n(d_\mathrm{eq}, z_{il})\, dd_\mathrm{eq}}{\sum_{i \in \mathcal{Z}} \Delta z_{il} \int_0^{+\infty} d_\mathrm{eq}^3 n(d_\mathrm{eq}, z_{il})\, dd_\mathrm{eq}} \tag{A10}$$

$$= \frac{\sum_{i \in \mathcal{Z}} \Delta z_{il} \left[ \frac{4^4}{6} \mathrm{IWC}_{il}^5 / N_{0,il}^* \left( \frac{6}{\pi \rho_\mathrm{w}} \right)^5 \right]^{1/4}}{\sum_{i \in \mathcal{Z}} \Delta z_{il} \frac{6}{\pi \rho_\mathrm{w}} \mathrm{IWC}_{il}} \tag{A11}$$

$$= \frac{4}{(\pi \rho_\mathrm{w})^{1/4}} \frac{\sum_{i \in \mathcal{Z}} \Delta z_{il} \left( \mathrm{IWC}_{il}^5 / N_{0,il}^* \right)^{1/4}}{\mathrm{IWP}_l}. \tag{A12}$$

Then, the $D_\mathrm{m}$ for a set of profiles indexed by $\mathcal{P}$ is

$$D_\mathrm{m} = \frac{\int_{z_0}^{z_\mathrm{max}} \int_0^{+\infty} d_\mathrm{eq}^4 n(d_\mathrm{eq}, z)\, dd_\mathrm{eq} dz}{\int_{z_0}^{z_\mathrm{max}} \int_0^{+\infty} d_\mathrm{eq}^3 n(d_\mathrm{eq}, z)\, dd_\mathrm{eq} dz} \tag{A13}$$

$$= \frac{\sum_{l \in \mathcal{P}} \sum_{i \in \mathcal{Z}} w_l \Delta z_{il} \int_0^{+\infty} d_\mathrm{eq}^4 n(d_\mathrm{eq}, z_{il})\, dd_\mathrm{eq}}{\sum_{k \in \mathcal{P}} \sum_{i \in \mathcal{Z}} w_k \Delta z_{ik} \int_0^{+\infty} d_\mathrm{eq}^3 n(d_\mathrm{eq}, z_{ik})\, dd_\mathrm{eq}} \tag{A14}$$

$$= \frac{\sum_{l \in \mathcal{P}} \sum_{i \in \mathcal{Z}} \Delta z_{il} \int_0^{+\infty} d_\mathrm{eq}^4 n(d_\mathrm{eq}, z_{il})\, dd_\mathrm{eq}}{\sum_{k \in \mathcal{P}} \sum_{i \in \mathcal{Z}} \Delta z_{ik} \int_0^{+\infty} d_\mathrm{eq}^3 n(d_\mathrm{eq}, z_{ik})\, dd_\mathrm{eq}} \cdot$$

$$\cdot \frac{\frac{\sum_{i \in \mathcal{Z}} \Delta z_{il} \int_0^{+\infty} d_\mathrm{eq}^3 n(d_\mathrm{eq}, z_{il})\, dd_\mathrm{eq}}{\sum_{i \in \mathcal{Z}} \Delta z_{il} \int_0^{+\infty} d_\mathrm{eq}^3 n(d_\mathrm{eq}, z_{il})\, dd_\mathrm{eq}}}{1} \tag{A15}$$


$$= \frac{\sum_{l \in \mathcal{P}} \sum_{i \in \mathcal{Z}} \Delta z_{il} \int_0^{+\infty} d_\mathrm{eq}^3 n(d_\mathrm{eq}, z_{il})\, dd_\mathrm{eq} D_{\mathrm{m},l}}{\sum_{k \in \mathcal{P}} \sum_{i \in \mathcal{Z}} \Delta z_{ik} \int_0^{+\infty} d_\mathrm{eq}^3 n(d_\mathrm{eq}, z_{ik})\, dd_\mathrm{eq}} \tag{A16}$$

$$= \frac{\sum_{l \in \mathcal{P}} \sum_{i \in \mathcal{Z}} \Delta z_{il} \mathrm{IWC}_{il} D_{\mathrm{m},l}}{\sum_{k \in \mathcal{P}} \sum_{i \in \mathcal{Z}} \Delta z_{ik} \mathrm{IWC}_{ik}} = \frac{\sum_{l \in \mathcal{P}} \mathrm{IWP}_l D_{\mathrm{m},l}}{\sum_{k \in \mathcal{P}} \mathrm{IWP}_k}. \tag{A17}$$

That is, it is also given by averaging the $D_m$ of each profile weighted by its IWP.



*Author contributions.* All authors contributed to the project through discussions. AA collected the data, performed the training and evaluation of the models, and prepared the manuscript, including all visual material. PE supervised the project and provided scientific feedback. SP

participated with the retrieval of $D_\mathrm{m}$ and machine learning feedback. SP also developed the quantnn Python package, available at https://github.com/simonpf/quantnn (last access 24 May 2022), which has been used to implement the retrievals.

*Competing interests.* The authors declare that they have no conflict of interest.

*Acknowledgements.* The contributions from PE and SP were covered by Swedish National Space Agency grant 154/19. The computations were performed on resources at Chalmers Centre for Computational Science and Engineering (C3SE) provided by the Swedish National

Infrastructure for Computing (SNIC). We would also like to thank the Satpy contributors that clarified the presence of a Meteosat georeferencing offset to us and how to handle it.



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
