# Peer review of "Ice water path retrievals from Meteosat-9 using quantile regression neural networks"

_Atmospheric Measurement Techniques, 2022_

## Referee Comment (RC1)

**General comments**

This paper proposes an IWP retrieval method using machine learning. Combining a quantile regression neural network (QRNN) and a convolutional neural network (CNN), the measurements from Meteosat-9 SEVIRI IR-only channels can be used to retrieve IWP. The retrieval using QRNN and CNN is encouraging because the results show good agreement with the DARDAR dataset. The paper is clearly written and the study is well explained. I believe the manuscript should be published in AMT but I have a major concern and some minor comments.

My main concern is that I don't see any statistics and analysis results for the collocation database. More statistical plots of the retrieval database are suggested to provide. Since the collocation database essentially represents the prior knowledge about the ice cloud distribution, these plots could help to verify if the collocation database captures the right statistics.

**Specific comments**

Line 75: In general, the TIR band does not penetrate as well as microwave band and is only sensitive to signals from cloud tops, which means that for large IWPs, the IR-only measurement is likely to lack valid IWP information, especially when compared to the 94 GHz CPR of CloudSat. Therefore, the question is, if the range of IWP is not limited, are the IWPs

in the results that do not fall within the sensitive range of the IR band inferred from a priori information rather than from the measurement? Are such results reliable? Also, are the results better without constraining the range of IWP than constraining the range of IWP?

Line 130: How many DARDAR observations will there be in a SEVIRI cell in the best case? Figure 1 shows the DARDAR observations do not fill the SEVIRI cell due to the different resolution. How to illustrate that the DARDAR profile can represent the scene in the SEVIRI image?

Line 136: All data is randomly divided into training, validation and testing sets, which means that the features of the test set are also learned by the neural network. I think it is better to use a separate set of data such as data in 2012 for testing the generalization ability of the network.

Sec. 2.3: It is necessary to analyse the statistical characteristics of the distribution of the collocations with a plot. Although the relationships between IWP and visible and infrared (VISIR) radiances have been stated, it is not sufficiently visual. I suggest adding a plot here to illustrate the relationship between these parameters and IWP to show that the collocations are valid. What also needs to be illustrated is the coverage of the observations in the dataset and whether it is representative of the

majority scenarios. Also, what is the proportion of scenes with and without ice clouds in the dataset and is there a problem with uneven data distribution?

Line 181: What is the purpose of random image mirroring and rotation?

Line 250: Does the spatial information refer to the correlation between pixels? Why it is useful for retrieval?

Line 281: In the comparison between the two products, is the instantaneous IWP retrieval using the test dataset? Why not use the data for 2012 as used in the monthly mean diurnal cycles?

---

## Referee Comment (RC2)

The paper proposed by Adrià Amell and colleagues presents an inversion technique based on machine learning for the estimation of ice wather path (IWP) form Meteosat-9 observations with a focus on low latitudes. In their work, the authors both introduce and describe the topic with good details and discuss the potential and advantages of using artificial intelligence quantile-based regression methodologies over physics-based methods present in the literature.

In this context, the authors test various neural network architectures and compare the use of observations in the thermal infrared (IR) and/or visible bands as inputs. Finally, authors conclude that the architecture based on convolutional neural networks (CNNs) in which spatial information is integrated is the architecture that performs better, using, moreover, only observations in the infrared band as input. The presented approach offers several advantages over traditional methods, such as the ability to calculate diurnal cycles, a problem that for example CloudSat cannot solve due to its limited temporal and spatial sampling. Then, since the methodology is quantile based, it allows the developed methodology to obtain directly and in an integrated way an estimate of the uncertainty of the regressions.

The authors validated their work using CLASS that is thoroughly validated dataset based on traditional approaches. The obtained retrievals compare favourably with IWP retrievals in CLAAS. In my opinion, this last result arguably demonstrates the potential of this methodology highlight the possibilities to overcome limitations from physics-based approaches as demonstrated in other works recently published in literature Holl et al. (2014), Islam and Srivastava (2015) and Mastro et al. (2022).

However, in my opinion, some shortcomings are present in the paper framework that require a major review.

1) In section 3.2 authors describe the Network architecture and specifically they discuss the multilayer percepton (MLP) and the CNN configurations indicating their structural hyperparameters. I would argue that it is essential to describe in more detail this information and how the choice of these configurations was made. For example, for the MLP configuration, the authors indicate an architecture consisting of 16 hidden layers each composed of 128 hidden units assuming that it is the setup that achieves the best performance. How did they reach this finding? Has a tuning framework been used? If so, how was the hyperparameter space configured from which to begin the search for the best configuration? Also, were configurations with fewer hidden layers explored?

2) The authors indicate that Table 2 shows the input characteristics used by the analyzed architectures. I believe that as presented, the table does not make it easy to understand which of the inputs shown are used of the architectures presented. I understand that various configurations of inputs were used for each architecture. Anyway, I sugges the authors reformulate more clearly the information in Table 2 and contextualize it better.

3) In section 3.3 the authors discuss the training of the proposed configurations. Here they also introduce information regarding the inputs used. In general as presented the section is very confusing and a possible reader might find it difficult to read. I propose to move the choice of inputs to section 3.2 following the corrections of Table 2 indicated previously and to focus section 3.3 in providing details concerning only the training phase. In addition, a useful piece of information would be to show the learning curves (for each epoch of training and

validation) of the two configurations in order to demonstrate the absence of overfitting and underfitting problems.

4) Figure 4 shows the CNN architecture and in my opinion it is a bit misleading. I would like to propose to the authors to change the position of the DXception and Xception blocks next to the blocks themselves, because as they look they appear to be part of the input and output blocks.

---

## Author Comment (AC1)

**Responses to the comments from Anyonymous Referee 1 Preprint https://doi.org/10.5194/amt-2022-184**

Adrià Amell*    Patrick Eriksson    Simon Pfreundschuh

24 August 2022

The original text from the Anonymous Referee is presented in colour grey and our responses in black. At the end there is a section specifying the changes to the manuscript after reading and answering each comment.

*amell@chalmers.se

**Response to comments from Anonymous Referee #1**

**General comments**

This paper proposes an IWP retrieval method using machine learning. Combining a quantile regression neural network (QRNN) and a convolutional neural network (CNN), the measurements from Meteosat-9 SEVIRI IR-only channels can be used to retrieve IWP. The retrieval using QRNN and CNN is encouraging because the results show good agreement with the DARDAR dataset. The paper is clearly written and the study is well explained. I believe the manuscript should be published in AMT but I have a major concern and some minor comments.

My main concern is that I don't see any statistics and analysis results for the collocation database. More statistical plots of the retrieval database are suggested to provide. Since the collocation database essentially represents the prior knowledge about the ice cloud distribution, these plots could help to verify if the collocation database captures the right statistics.

We thank the referee for the nice and concise summary of the paper, as well as for the comments below. We address the major concern through the answers to the specific comments.

**Specific comments**

1. Line 75: In general, the TIR band does not penetrate as well as microwave band and is only sensitive to signals from cloud tops, which means that for large IWPs, the IR-only measurement is likely to lack valid IWP information, especially when compared to the 94 GHz CPR of CloudSat. Therefore, the question is, if the range of IWP is not limited, are the IWPs in the results that do not fall within the sensitive range of the IR band inferred from a priori information rather than from the measurement? Are such results reliable? Also, are the results better without constraining the range of IWP than constraining the range of IWP?

   We break down the answer to this comment to address the three questions asked.

   - If the range of IWP is not limited, are the IWPS in the results that do not fall within the sensitive range of the IR band inferred from a priori information rather than from the measurement?

     Let $Q$ represent the distribution of the IWP in the training data, and $P$ the distribution that can be constructed from the models prediction. We consider here $Q$ as the prior information. We can see that the measurements are not inferred from a priori information but rather from the measurement, as the expected value of $P$ is very different from the expected value of $Q$ ($1.18 \times 10^{-1}\,\mathrm{kg\,m^{-2}}$), particularly for the largest IWP values, which we understand is the range where the referee asks for. We can see that from Figures 5 and 7, and, albeit the expected value of $Q$ is not reported, estimate it from the distribution in Figure 10, which is very similar to $Q$.

     If this is not sufficient to clarify this doubt, we provide a mathematical-statistical argument. In Figure RC1.1 we show the Kullback-Leibler divergence between all predicted distributions $P$ from the test set, using the IR-only CNN network, and $Q$. The Kullback-Leibler divergence is a statistical distance that measures how different $P$ is from a reference distribution $Q$, denoted as $D(P \parallel Q)$. The larger its value, the more different the distributions are. Two distributions are equal if their divergence is 0. It is explained in the next paragraph how we numerically computed it. We can see that it tends to not be zero for any IWP, from which we can conclude that IWPs

in the results are inferred from the measurement rather than a priori information, as the predicted $P$ is different from $Q$.

In the general case, the Kullback-Leibler divergence for two probability measures $P$ and $Q$ is defined as

$$D(P \parallel Q) = \int_{\Omega} \mathrm{d}P \log_2 \frac{\mathrm{d}P}{\mathrm{d}Q}$$

where $\Omega$ is the set over which $P$ and $Q$ measure, and requires that $P$ is absolutely continuous with respect to $Q$, indicated as $P \ll Q$. If $P$ and $Q$ are defined over the reals and have a density, then it can be expressed as

$$D(P \parallel Q) = \int_{\mathbb{R}} p(x) \log_2 \left[ \frac{p(x)}{q(x)} \right] \mathrm{d}x$$

where $p$ and $q$ are the respective density functions. We define $F_{X|\mathbf{Y}}(x) = P(X \leq x)$ and $F_X = Q(X \leq x)$, which implies that $P \ll Q$ for this problem. We avoid mathematical rigurosity to be able to provide an intuition of how different $P$ and $Q$ tend to be, and approximate $D(P \parallel Q)$ with the Riemann sum

$$\hat{D}(P \parallel Q) = \sum_{i=1}^{n} \Delta F_{X|\mathbf{Y}}^{(i)} \log_2 \left[ \frac{\Delta F_{X|\mathbf{Y}}^{(i)}}{\Delta F_X^{(i)}} \right]$$

where $\Delta F_{X|\mathbf{Y}}^{(i)} = F_{X|\mathbf{Y}}(x_i) - F_{X|\mathbf{Y}}(x_{i-1})$, analogously for $\Delta F_X^{(i)}$, $F_X$ is the empirical cumulative distribution (CDF) of the training data, $F_{X|\mathbf{Y}}$ is the continuous piecewise linear extension of the predicted distribution to provide values for any $x \in X$, and $X = \{x_0, x_1, \ldots, x_n\}$ are all IWP values in the training data set.

[Figure]

**Figure RC1.1:** The Kullback-Leibler computed as described in text, for the test set predictions with the CNN. The median curve shows the trend at the local DARDAR IWP, and the quantiles indicate the dispersion at the local IWP. Linear scale until $1 \times 10^{-3}\,\mathrm{kg\,m^{-2}}$, and log scale afterwards.

- Are such results [that do not fall in the sensitive range of the IR band] reliable?

We can interpret the question in two ways. The first one is whether the quantile for values outside the sensitive range of the IR band are good estimates. This would require predicting another type of uncertainty in the problem, the uncertainty of the predicted quantiles. How to estimate this uncertainty when predicting quantiles (combining epistemic uncertainty with aleatoric uncertainty) is yet to be studied for retrievals, to the best of our knowledge. This is not discussed in the paper as we consider it to be completely out of the scope of this work.

The second interpretation we make is whether the predicted distribution $P$ captures better the reference value than the prior $Q$. We can see that the results are reliable because, on the test set, the expected value of $P$ tends to follow the identity line, although for the largest values the curve starts to deviate. It is sensible to assume this range (large IWPs) is where the machine cannot leverage differences between the IR measurements, if there are any. Note that, in both training and test set, values above $10 \, \mathrm{kg} \, \mathrm{m}^{-2}$ (the right limit of the abscissa in Figures 5 and 7) represent less than 1.6% of the values. Thus, we expect IWP above this threshold to be very rare for the given region of interest.

- Are the results better without constraining the range of IWP than constraining the range of IWP?

We make two interpretations of this question: on the one hand, that the results are evaluated on a constrained range of IWP on the networks trained on all IWP data, and on the other hand that the networks are trained and evaluated on a constrained IWP range. In the former case, what we call summary statistics (RMSE, MAE, BIAS, CRPS, $r_\mathrm{S}$) will change. For instance, if in the test set only IWP $< 1 \, \mathrm{kg} \, \mathrm{m}^{-2}$ values are considered, the CNN summary statistics presented in Figure 7 will change to `RMSE = 2.09E-1`, `MAE = 4.03E-2`, `bias = 3.04E-2`, `CRPS`$_\mu$`= 1.84E-2`, `CRPS`$_\mathrm{m}$`= 5.04E-4`, all in $\mathrm{kg} \, \mathrm{m}^{-2}$, and `r`$_\mathrm{S}$`= 0.85`. Except the bias and $r_\mathrm{S}$ which are worse, although $r_\mathrm{S}$ is almost equal, the other parameters show better results. This is due to the influence that large values can have over small values in the summary statistics, and can be misleading to evaluate results only on these parameters (line 212). This is the reason why such parameters should be computed on the same set of values when comparing the results of different methods, as done in the paper. Concerning the graphical evaluation, as in Figures 5 and 7, then it is only necessary to look at the desired range.

If we trained the networks on a constrained range of IWP, then the network would have a different a priori information. While it could achieve better results when evaluating against IWP in the constrained range, it raises two questions. Firstly, how it would perform for IWP values outside the constrained range. Secondly, if values outside the range are to be excluded from the retrieval, as they are not values which the network has been exposed during training, the retrieval algorithm should have a way to identify them. One could also argue in favour or against the values for the limits of the constrained range, which could be considered arbitrary. For all these reasons, we think it is better to train using the full range of IWP for retrievals under any circumstances. Nevertheless, we show in Figure RC1.2 the results of training and evaluating the MLP network corresponding to Figure 7 with constrained ranges, where it is seen that training under constrained ranges is actually worse, based on

the curves presented (the best of five trainings for each range is presented). However, this MLP was tuned to have a good performance for any IWP range, and not the contrained ranges used.

[Figure]

**Figure RC1.2:** Results on the test set for the MLP network corresponding to Figure 7, but trained and evaluated on the constrained range of IWP indicated by each subplot title.

2. Line 130: How many DARDAR observations will there be in a SEVIRI cell in the best case? Figure 1 shows the DARDAR observations do not fill the SEVIRI cell due to the different resolution. How to illustrate that the DARDAR profile can represent the scene in the SEVIRI image?

We first clarify what we mean with DARDAR observation in this answer. Using Figure 1, we refer to a DARDAR observation as either the original (blue + symbol) or replicated (orange + symbol) DARDAR profile. In the best case, there are 12 DARDAR observations per pixel. However, this correspond to edges of the region of interest, predominantly in the lower right corner. This is a consequence of the projection used, as it is the furthest area from the centre of the projection. However, an analysis of Figure RC1.3 shows that 9 profiles is probably a better answer, as this is seen uniformly throughout the region of interest (ROI).

In the second question of this comment, we understand the word "scene" (and consequentially the question) as, "given a pixel $P$ in the SEVIRI image, how can collocated DARDAR profiles in $P$ represent the complete IWP in $P$, given that they can only partially cover

*P*?". We argue that this is a problem that does not have a good solution, as one could set a threshold on the numbers of profiles required to represent a "scene". However, we consider any threshold value completely arbitrary. Therefore, to not lose any information, we considered all pixels with collocations. In the worst collocation case, where there is only one collocated profile in *P* completely irrepresentative of the "scene", the networks will ideally interpret it as noise, as long as such case is very rare in the training data. We also suggest (in line 403) that this concern can be addressed in further research by incorporating the disagreement between profiles in a pixel *P* during training.

[Figure]

**Figure RC1.3:** Each dot represents a SEVIRI pixel (not to scale) in the ROI. Each subplot title indicates how many DARDAR profiles, as described in text, are located in the SEVIRI pixel and therefore were used to obtain the collocated value for the pixel. The subplot title also indicates how often the given number of profiles are found in a SEVIRI pixel. All numbers computed from the training set data.

3. Line 136: All data is randomly divided into training, validation and testing sets, which means that the features of the test set are also learned by the neural network. I think it is better to use a separate set of data such as data in 2012 for testing the generalization ability of the network.

We understand the concerns raised by using a random division of the data available for testing the generalization ability instead of using, for example, a temporal division, as the referee thinks. We interpret the concerns as that neighbour image samples, which can be allocated to different sets, can correspond to similar or related atmospheric states. We

think it is a sensible point. However we made this choice motivated by the data available and that no information is re-used between the training and (validation and) test sets. Note that the image samples do not overlap. We elaborate below.

The collocations database consist of nearly 3 years of day and night collocations. The upper time bound is due to that day and night collocations are only available before April 2011 because of the CloudSat battery anomaly (line 96). The lower time bound is due to the files provided by the EUMETSAT Data Store: files before 6 May 2008 were processed with another algorithm version than those posterior to this date. We have not found a citable reference that clearly explains this. This information is scattered on the Internet and can be noticed from the SEVIRI file names. Therefore, our database contains nearly all possible day and night collocations for SEVIRI images processed with the same algorithm version. That is, using data in 2012 is not suitable to test the overall generalization, as it lacks nighttime collocations. For completeness, we also state here "nearly all" instead of "all" as, at the time of finalizing the manuscript, we noticed that relatively very few random DARDAR files had not been downloaded; we cannot foresee how re-doing all steps and analyses with "all" collocations would bring any benefit, and in our opinion it would only have consumed a substantial amount of time.

One could argue to use, for example, 2009 and 2010 for training (and validation), and the remaining data (from 2008 and 2011) for testing. Figure RC1.4 shows that such strategy would miss to test the generalization ability for certain time ranges (or even train it), and can even miss valuable information in the training. If several more years of data were available, then a train-test split based on years would be fair. A more strategic approach with our database would be to differentiate between orbits, that is no samples from the same orbit should be in training and test sets. However, this approach should ensure that all areas are well represented in the different sets: given an area, how to handle if, in two consecutive overpasses at the same time of day (which are 16 days apart), assigned to the training and test sets, respectively, the training samples capture a common state, but the test set samples an unusual one? More convolved approaches could be thought, but that does not imply they test better the generalization ability.

[Figure]

**Figure RC1.4:** Samples (32×32 pixels images) in the training, validation and test sets aggregated.

4. Sec. 2.3: It is necessary to analyse the statistical characteristics of the distribution of the collocations with a plot. Although the relationships between IWP and visible and infrared (VISIR) radiances have been stated, it is not sufficiently visual. I suggest adding a plot here to illustrate the relationship between these parameters and IWP to show that the collocations are valid. What also needs to be illustrated is the coverage of the observations in the dataset and whether it is representative of the majority scenarios. Also, what is the proportion of scenes with and without ice clouds in the dataset and is there a problem with uneven data distribution?

Figure RC1.5 provides the plot requested by the referee, although for the daytime observations. If the same plot is produced using daytime and nighttime data, or nighttime data only, then the plots for channels 1-3, and to a lesser extent 4, change abruptly, as these channels depend on a solar contribution; there are small diferences between the thermal infrared channels between day and night, which we believe can be caused by the diurnal cycle of atmospheric ice itself. It can be observed that the distributions are reasonable: for infrared channels, the colder a SEVIRI pixel is, the higher the IWP; for the solar channels, the higher the reflectance is, the higher the IWP.

[Figure]

**Figure RC1.5:** Distributions of the SEVIRI channels plotted against the DARDAR IWP, daytime data only.

Concerning the request to illustrate the observations coverage, it is partially answered by Figure RC1.4 and complemented by Figure RC1.6, which will replace Figure 2 in the paper.

[Figure]

**Figure RC1.6:** Figure 2 updated, with the DARDAR profiles (not to scale).

Regarding the last question from the Anonymous Referee, we can understand the word "scene" in the question "what is the proportion of scenes with and without ice clouds in the dataset" as either a SEVIRI pixel (case A) or a "sample" (case B), the word we used in our paper to describe the 32×32 pixels images used. DARDAR provides a flag for each vertical bin, describing its content. Among these flags, exist the values `ice`, `ice + supercooled`, `liquid warm`, and `supercooled`. We tagged profiles with any of these flags present as cloudy. In case A, nearly 78% of pixels have a cloud, of which 72% have the flag `ice`, and thus are ice clouds (the same result is obtained if `ice + supercooled` is also considered). That is, 56% of the pixels have an ice cloud. These numbers change for case B, as 128 pixels are included per sample. 97% of the samples have pixels with cloud flags, of which 70% have ice clouds, which implies that 68% of the samples have ice clouds.

Finally, the problem with an uneven data distribution is that the most challenging range to retrieve IWP from the thermal IR channels has limited data. This implies that, even if it would be physically possible to perform excellent retrievals of large IWP from IR radiances, any machine learning model would struggle as there is not much data to learn from. For example, in the training set there are only 374 pixels with IWP $> 10\,\mathrm{kg\,m^{-2}}$ compared to more than $13\,000$ pixels in the range IWP $\in [10^{-1}, 1]\,\mathrm{kg\,m^{-2}}$. In this particular case, techniques to combat uneven data distributions, such as oversampling, would alter any prior information for the retrievals. We then argue that, while this might produce better results for the oversampled range, it may worsen the retrievals for other ranges if such a technique is not used carefully, and are skeptical whether that would bring a substantial general retrieval improvement. We refer to the answer of the specific comment #1 that complements this comment.

5. Line 181: What is the purpose of random image mirroring and rotation?

Better generalization. Note that rotating multiples of 90 degrees does not alter the pixel information (no resampling has to be done, which would happen for non-multiples of 90 degrees). With this technique, the samples are presented differently to the convolutional filters, which have no information about whether the samples had any transformation, but

the network has to retrieve the same values for each input value. Therefore, these filters are trained on data that is presented differently in each epoch, helping the network be better prepared for inference with non-training data.

6. Line 250: Does the spatial information refer to the correlation between pixels? Why it is useful for retrieval?

It corresponds to exploiting the correlation between neighbouring pixels by using the CNN. We state it is useful from the results, which are summarized also in line 250. Intuitively, making good use of information surrounding an image pixel (the neighbouring pixels) should help to constrain better the retrieval and obtain a better performance.

7. Line 281: In the comparison between the two products, is the instantaneous IWP retrieval using the test dataset? Why not use the data for 2012 as used in the monthly mean diurnal cycles?

Yes, it uses the test dataset (with few SEVIRI samples excluded as those observations are not present in CLAAS, line 285). The reason for using the test set is to not introduce yet another data set, such that, if it is desired, one can compare with the results from the other sections where the test set is used. Note that the diurnal cycles do not have DARDAR data, they only consist of the CLAAS diurnal cycles and the predictions of the CNN using IR-only channels.

**Manuscript changes after the comments from Anonymous Referee #1**

- Fig. RC1.6 replaces Fig. 2, and updated caption accordingly
- Included the percentage of ice cloud "scenes" (pixels and samples).
- Created a section in the supplementary material, which is referred to in the paper if a reader wants more details, with an analysis and statistics of the collocation database, with:
  - More explicit explanation of the coverage of the collocations, covering why the lower time bound is 6 May 2008.
  - Figures RC1.4, RC1.3 to detail the temporal coverage of the collocations and how many profiles are used per collocation, repectively.
  - Figures describing the percentage of cloudy "scenes" (pixels and samples).
  - Relationships between the collocated DARDAR IWP and the visibile and infrared radiances, separated by daytime and nighttime data (this includes Fig. RC1.5).
  - A table with several quantiles and mean value of the different data sets used in the paper.
- Split the line where it is mentioned that random rotation and mirroring is used, to clarify that it is done on each data access, and mentioned that it is to have a better generalization.

Concerning the Kullback-Leibler analysis presented as a second answer to comment #1 (in the form of a mathematical-statistical argument), we do not plan to update the manuscript with this nor include it in the supplementary material. We think that it would only distract the average reader, and that the first answer we provide to comment #1 (large IWPs are inferred from the measurements as they do not match the expected value of the a priori information) is the general understanding.

---

## Author Comment (AC2)

**Responses to the comments from Anyonymous Referee 2 Preprint `https://doi.org/10.5194/amt-2022-184`**

Adrià Amell*      Patrick Eriksson      Simon Pfreundschuh

24 August 2022

The original text from the Anonymous Referee is presented in colour grey and our responses in black. At the end there is a section specifying the changes to the manuscript after reading and answering each comment.

*amell@chalmers.se

**Response to comments from Anonymous Referee #2**

The paper proposed by Adrià Amell and colleagues presents an inversion technique based on machine learning for the estimation of ice wather path (IWP) form Meteosat-9 observations with a focus on low latitudes. In their work, the authors both introduce and describe the topic with good details and discuss the potential and advantages of using artificial intelligence quantile-based regression methodologies over physics-based methods present in the literature.

In this context, the authors test various neural network architectures and compare the use of observations in the thermal infrared (IR) and/or visible bands as inputs. Finally, authors conclude that the architecture based on convolutional neural networks (CNNs) in which spatial information is integrated is the architecture that performs better, using, moreover, only observations in the infrared band as input. The presented approach offers several advantages over traditional methods, such as the ability to calculate diurnal cycles, a problem that for example CloudSat cannot solve due to its limited temporal and spatial sampling. Then, since the methodology is quantile based, it allows the developed methodology to obtain directly and in an integrated way an estimate of the uncertainty of the regressions.

The authors validated their work using CLASS that is thoroughly validated dataset based on traditional approaches. The obtained retrievals compare favourably with IWP retrievals in CLAAS. In my opinion, this last result arguably demonstrates the potential of this methodology highlight the possibilities to overcome limitations from physics-based approaches as demonstrated in other works recently published in literature Holl et al. (2014), Islam and Srivastava (2015) and Mastro et al. (2022).

We thank the referee for the nice summary of the paper, as well as for the comments and suggestions below.

However, in my opinion, some shortcomings are present in the paper framework that require a major review.

1. In section 3.2 authors describe the Network architecture and specifically they discuss the multilayer percepton (MLP) and the CNN configurations indicating their structural hyperparameters. I would argue that it is essential to describe in more detail this information and how the choice of these configurations was made. For example, for the MLP configuration, the authors indicate an architecture consisting of 16 hidden layers each composed of 128 hidden units assuming that it is the setup that achieves the best performance. How did they reach this finding? Has a tuning framework been used? If so, how was the hyperparameter space configured from which to begin the search for the best configuration? Also, were configurations with fewer hidden layers explored?

   We understand that it can be concerning to not provide more information about the design choices, particularly for those readers who are more knowledgeable in machine learning. We want the reader to be focused on the retrieval performance with machine learning, and not distracted by details of the choices in the machine learning models. In the three works cited that use neural networks to retrieve IWP, no description is given for the network choice (Holl et al., 2014), a rule of thumb is used without any hyperparameter search employed (Islam and Srivastava, 2015) or, the most recent work, only states what framework was used but not how it was configured (Mastro et al., 2022). Despite the level of detail clearly differs, we are updating the paper with some information at a very high-level. Nevertheless we answer the questions in detail here, as the hyperparameter tuning represented a substantial amount of work.

A subset $\mathcal{S}_t$ of the training data set $\mathcal{S}$ was used to train different models, which are described in the next two paragraphs. Another subset $\mathcal{S}_v \subset \mathcal{S}$, $\mathcal{S}_v \cap \mathcal{S}_t = \emptyset$ was used to evaluate the performance of each model. The performance on $\mathcal{S}_v$ determined our model choice. All models were trained and evaluated several times to make solid choices. No software tuning framework was used for this step. Instead, the (human) evaluation explained in the manuscript (line 175) was used for this step.

The CNN configuration and choices comes from experience and expertise acquired through previous or parallel projects from the authors. All these projects consists of retrievals with QRNNs. Furthermore, choices for the CNN are imposed by data and hardware constraints. In using a similar network as in those other projects we were able to adapt existing code for this work. This reduced the chances of running into silent bugs or inappropriately using the ML libraries. We find that presenting the choices for the blocks in the CNN model itself would require a paper of its own, since we also evaluated small changes in the CNN choices, which did not present remarkable benefits in this retrieval problem. The number of Xceptions blocks $n$ was searched thoroughly over the grid $n = \{0, 1, 2, 4\}$, and the number of filters in $k = \{64, 128\}$. We observed that with $n = 2$ and $k = 128$ the performance tended to be better, yet only marginally better than the other choices in several repetitions of this hyperparameter exploration. Therefore, we did not explore deeper for a better final configuration.

Concerning the MLP, the goal was to have a simple MLP without any elaborated design choice as a benchmark base for any CNN network. This also made it computationaly cheaper to train and define a hyperparameter space. In this case, we explored configurations with $l = \{1, 2, 4, 8, 16, 32, 64, 128, 256\}$ hidden layers, $n = \{8, 16, 32, 64, 128, 256\}$ hidden neurons, and the three input settings used. Through the manual evaluation, we observed that $l = 16$ and $n = 128$ tended to perform equal or better than the other choices in several repetitions of the hyperparameter exploration. More details could be given on the MLP performance for this problem, but as in the CNN case, we also think this would deserve a paper of its own: only determining the MLP network candidate to then train with $\mathcal{S}$ represents $|l| \times |n| \times (3 \text{ input settings}) = 162$ options to compare, but since more than one training execution was performed for each hyperparameter configuration, there would be many more results to discuss.

2. The authors indicate that Table 2 shows the input characteristics used by the analyzed architectures. I believe that as presented, the table does not make it easy to understand which of the inputs shown are used of the architectures presented. I understand that various configurations of inputs were used for each architecture. Anyway, I sugges the authors reformulate more clearly the information in Table 2 and contextualize it better.

We think that the information in Table 2 is clear, but we are updating the Table caption sentence "Input features used." to "Input features used for each input settings.". Throughout the text, the input settings for each network are indicated whenever they cannot be determined from the context, whether they are contained in the Figure (as in Figure 5), in the Figure caption (Figure 7, Figure 9), or in the text itself, for example, lines 247 or 259. This also implies that various configurations of inputs were used only for the MLP, as described in Sect. 4.1, and the rest of results, including the features used for the CNN, build only on using the features referred as IR input settings, as the first line (line 247) of Sect. 4.2 states and the following (sub)sections remind.

3. In section 3.3 the authors discuss the training of the proposed configurations. Here they also introduce information regarding the inputs used. In general as presented the section is very confusing and a possible reader might find it difficult to read. I propose to move the choice of inputs to section 3.2 following the corrections of Table 2 indicated previously and to focus section 3.3 in providing details concerning only the training phase. In addition, a useful piece of information would be to show the learning curves (for each epoch of training and validation) of the two configurations in order to demonstrate the absence of overfitting and underfitting problems.

We appreciate this observation and are moving the text accordingly. We would also like to remark that we did observe overfitting, which is why early stopping on the validation loss was used (line 178). Therefore, we do not see any added value in adding the learning curves of the seven networks presented in the paper, but we will provide Figure RC2.1 as supplementary material. For comparison with the three works cited that use neural networks to retrieve IWP, only Mastro et al. (2022) present a learning curve, but only for one of the networks therein (cf. comment #1). As a side note, we identified that the bursts in the loss function are related to the behaviour of the chosen optimizer (and its hyperparameters) in the landscape of the loss function for this problem, but we consider any further discussion and analysis on this regard out of the scope of the paper.

[Figure]

**Figure RC2.1:** Learning curves, with the early stopping epochs indicated.

4. Figure 4 shows the CNN architecture and in my opinion it is a bit misleading. I would like to propose to the authors to change the position of the DXception and Xception blocks next to the blocks themselves, because as they look they appear to be part of the input and output blocks.

Figure 4 is designed such that it fits nicely in a two-column paper, and the suggestion does not fit our intention. We understand that it can be misleading. Figure RC2.2 will replace the current Figure in the paper (the colour choice is arbitrary).

[Figure]

**Figure RC2.2:** Diagram that will replace Figure 4.

**Manuscript changes after the comments from Anonymous Referee #2**

- Added a high-level info on how the hyperparemeters for the networks were selected in Sect. 3.2.

- Table 2 caption from "Input features used." to "Input features used for each input settings.".

- Move text according to referee comment #3.

- All learning curves from all networks presented in the paper (Fig. RC2.1) added as supplementary material.

- Figure 4 replaced by Fig. RC2.2.

---

## Author Comment (AC3)

**Responses to the comments from Anyonymous Referee 3 Preprint `https://doi.org/10.5194/amt-2022-184`**

Adrià Amell*      Patrick Eriksson      Simon Pfreundschuh

24 August 2022

The original text from the Anonymous Referee is presented in colour grey and our responses in black. At the end there is a section specifying the changes to the manuscript after reading and answering each comment.

*amell@chalmers.se

**Response to comments from Anonymous Referee #3**

The paper, "Ice water path retrievals from Meteosat-9 using quantile regression neural networks," develops a new approach toward estimating cloud ice water path during any time of the day using a machine learning method. The technique is trained with matched SEVIRI pixels and DARDAR profiles using a quantile regression neural network that permits an estimate of the uncertainty for each retrieval. Two versions are applied, a single pixel method (MLP) and a single pixel plus surrounding pixel data, a convolutional neural net (CNN). The latter was found to be superior to the former in that the average uncertainty was reduced, although the CNN tends to "smear" the IWP signal across neighboring pixels. Three input datasets were tested: VISIR (daytime) using all but one SEVIRI channel, an infrared (IR) only method using all non-solar SEVIRI channel, and a subset IR case using only two channels to simulate an historical Meteosat imager. All were trained using DARDAR data and their results were compared to DARDAR data taken within the same time frame as the training set. The VISIR performed best, but is limited to daytime and would require training for various solar angles that are not available for the DARDAR. The IR input produced quite acceptable results that are consistent in relative terms with the daytime portions of diurnal cycles of IWP determined from other passive sensor methods. The IR subset input shows less skill but provides information that is not obtainable with more physically based retrievals. This approach shows promise for improving the estimation of IWP at all times of day.

We thank the referee for the nice summary of the paper, as well as for the comments and suggestions below.

I recommend publication with a few revisions.

1. It would help in section 2.3 to use the same terminology in the text and Figure 1 description of the collocation. "Cell" is only mentioned in the caption, not in the text. SEVIRI "pixel" is used in the text. Also, the caption should note the units used for lat and lon, as degree is the usual unit. Is it correct to assume that the SEVIRI pixels used for given image were at least 16 km apart?

   The assumption is not correct. We believe this question comes from confusing the limits of the abscissa in the plot with the size of a SEVIRI pixel in the projection used, delimited by the green lines. We are replacing "cell" with "pixel" in Figure 1, as suggested. This should reduce the risk of assuming that pixels are at least 16 km apart. We will also indicate in the Figure 1 caption that the units for the coordinates are kilometres.

2. The units of the statistical parameters in Figure 5 and 7 are given in kg m-3. That is good. But, the mean DARDAR value should be noted for each plot, or the values given in percent of the mean DARDAR value in the text.

   We are adding in the captions of Figure 5 and 7 the DARDAR mean of the observations in the test set.

3. Line 280: By stating that "CLAAS has been thoroughly validated" suggests that the CLAAS IWP values agreed well with actual IWP measurements. The cited studies showed that CLAAS agreed well with similar passive remote sensing techniques, but not particularly well with DARDAR, the "ground truth" used here. For the most part, the CLAAS values are significantly lower than their DARDAR counterparts, as indicated later in the discussion. As the results shown in the citations vary, and none arise from an actual comparison with any in situ data, "thoroughly validated" is a bit of an overstatement. I would suggest rewriting this line, so that it is no surprise to find the CLAAS mean running below that from the CNN in Fig. 9.

Thank you for this remark. Our original intention with the expression "CLAAS has been thoroughly validated" is to show that this dataset has works that compared IWP retrievals in the CLAAS dataset with other sources. We are updating the paragraph that starts that line to avoid this surprise, as well as removing "thoroughly validated" in the abstract, for consistency.

4. Figure 10. This plot is difficult to examine closely. I think it would be easier to compare the two methods by putting them on the same graph with the two scales, and maybe only using 4 months instead of all twelve, just to illustrate the relative consistency.

We assume the referee meant Figure 11 instead of 10. We have prepared Figure RC3.1 to replace Figure 11, and we are updating the text accordingly. Note that the choice of dashed and solid lines for the CLAAS dataset and CNN retrievals, respectively, and the colour choice for each month is only for the best clarity in the plot. For the curious reader, we are also including the current Figure 10 in supplementary material.

[Figure]

**Figure RC3.1:** Diurnal cycles for four arbitrary months in 2012. Note the different vertical ranges. The default matplotlib (Python library for plotting) method for determining the vertical limits was used. This Figure will replace Fig 10.

5. Mean mass height (Zm) and mean mass size (Dm) are derived with somewhat mixed results. It would be helpful if the authors could remind us of the importance of these parameters.

Ice water path (IWP) is an integrated value of the ice water content (IWC), but neither can we have any information about at what height IWC is located nor about the size of the ice crystals constituting the IWC. Estimating $Z_m$ and $D_m$ gives some information about these two problems and, therefore, can help to characterize better atmospheric ice. We are adding to the paper that these parameters help characterize better atmospheric ice.

**Manuscript changes after the comments from Anonymous Referee #3**

- Figure 1: "pixel" replaces "cell", and coordinate units remarked in the caption.

- Added mean of DARDAR IWP in the test set in Figs. 5 and 7. Also, added "daytime" to "test data" in Fig. 5, to clearly indicate that no nighttime observations were used there.

- Replaced the expression "CLAAS has been thoroughly validated" with "CLAAS IWP has been analyzed against DARDAR and compared with MODIS retrievals", and removed "thoroughly validated" from the abstract sentence "[in CLAAS], a thoroughly validated dataset based on a traditional approach.".

- Fig. RC3.1 replaces Fig 11, and Fig. 11 is moved to supplementary material. Text and captions adapted to match that four months are presented in the main text.

- Motivated the the importance of $Z_m$ and $D_m$ in the beginning of section 4.3, before presenting their retrieval results.